# Regression under demographic parity constraints via unlabeled post-processing

**Gayane Taturyan**
IRT SystemX, Université Gustave Eiffel,
Université Paul-Sabatier
`gayane.taturyan@univ-eiffel.fr`

**Evgenii Chzhen**
CNRS, Université Paris-Saclay
`evgenii.chzhen@cnrs.fr`

**Mohamed Hebiri**
Université Gustave Eiffel
`mohamed.hebiri@univ-eiffel.fr`

## Abstract

We address the problem of performing regression while ensuring demographic parity, even without access to sensitive attributes during inference. We present a general-purpose post-processing algorithm that, using accurate estimates of the regression function and a sensitive attribute predictor, generates predictions that meet the demographic parity constraint. Our method involves discretization and stochastic minimization of a smooth convex function. It is suitable for online post-processing and multi-class classification tasks only involving unlabeled data for the post-processing. Unlike prior methods, our approach is fully theory-driven. We require precise control over the gradient norm of the convex function, and thus, we rely on more advanced techniques than standard stochastic gradient descent. Our algorithm is backed by finite-sample analysis and post-processing bounds, with experimental results validating our theoretical findings.

## 1 Introduction

Algorithmic fairness is an umbrella term for a subset of machine learning research that aims to better understand, quantify, mitigate, evaluate, and conceptualize negative and/or positive effects of data-driven algorithms on the society. At least one direction in this field falls within theoretical machine learning, where a form of fairness constraint, mainly inspired by common sense and formalized within mathematical framework, is proposed as an arguably reasonable proxy for a definition of ethical and non-discriminatory prediction. Even more particular sub-field of this research direction is formalized within a paradigm of group fairness, that aims at mitigating negative impact (or provide equal treatment to) towards sub-populations that share a common sensitive characteristic. Many works fall within this category (Barocas et al., 2018, Calders et al., 2009, Chiappa et al., 2020, Dwork et al., 2011, Feldman et al., 2015, Gordaliza et al., 2019, Hardt et al., 2016, Jiang et al., 2020, Lum and Johndrow, 2016, Zafar et al., 2017, Zemel et al., 2013, just to name a few).

Even without going into debates on the relevance of a given definition of fairness, many, purely mathematical and algorithmic questions remain unanswered in this field. The best theoretical understanding of the problem is available for the demographic parity constraint in case of *awareness*—the situation when the sensitive attribute is available at inference time (Agarwal et al., 2019, Chiappa et al., 2020, Chzhen and Schreuder, 2020b, Chzhen et al., 2019, Denis et al., 2024, Gaucher et al., 2023, Le Gouic et al., 2020). The latter case is well studies both in classification and regression setups. This is no longer the case for other fairness constraints or the *unawareness* setup—the situation when the sensitive attribute is not available at inference time. In particular, while the case of

38th Conference on Neural Information Processing Systems (NeurIPS 2024).

classification has been studied before from algorithmic and mathematical perspectives (Chzhen et al., 2019, Gaucher et al., 2023, Gordaliza et al., 2019, Hardt et al., 2016), the regression setup remains largely under explored and many methods lack strong theoretical evidences. In particular, to date, none of previous works effectively build computationally-efficient, fully theory-driven algorithm for the problem of regression under the demographic parity constraint in the case of unawareness. The present work fills this gap. Relying on previous ideas of discretization that goes back to Agarwal et al. (2019), we design a smooth convex objective function whose exact solution yields a fair and optimal prediction function. It turns out that this objective admits a first-order stochastic oracle that can be evaluated using only one independent sample of feature vector, thus allowing for stochastic optimization approach. Furthermore, despite the convexity, we show that the key quantity to control is the gradient (or rather a gradient-map) of this objective function, deviating from the more common setup of controlling the optimization error measured by the objective function. We deploy recent machinery of Allen-Zhu (2021) and Foster et al. (2019) that allows to achieve this goal, properly setting all the hyper-parameters and recovering the usual statistical rate $1/\sqrt{T}$ for both fairness and risk guarantees — $T$ being the number of samples.

Our work falls withing the realm of post-processing methods—another umbrella term that combines all the methods that perform a refitting of a base estimator to satisfy a certain constraint.

Importantly, due to the careful design of the above mentioned objective function, we can perform this post-processing in an online manner using a stream of i.i.d. *unlabeled* data without keeping it in memory, making it attractive in practice. Our approach is based on a combination of ideas from previous contributions to fairness from Agarwal et al. (2019) and Chzhen et al. (2020b) and recent stochastic optimization literature (Allen-Zhu, 2021, Foster et al., 2019) that deals with stationary point-type guarantees in the case of convex optimization.

**Contributions** Our contribution is three-fold: **i)** we significantly enhance the discretization strategy of Chzhen et al. (2020b) accommodating multiple sensitive features, relaxed fairness constraints, and unawareness setup; we introduce entropic regularization for this problem and design a dual convex objective from it; **ii)** we design a semi-supervised post-processing algorithm and show that it enjoys strong theoretical guarantees; **iii)** we perform numerical simulations demonstrating the relevance of our approach in practice.

**Organization.** This paper is organized as follows: in Section 2 we present the problem setup and introduce main problem-related notation; in Section 3 we describe our methodology step-by-step and highlight main challenges and relations to other results; in Section 4 we gives technical details of the proposed approach; Section 5 contains main theoretical results of the work; finally, Section 6 contains empirical evaluation of our method. All the proofs are postponed to the appendix.

**Notation.** Let us present generic notation that is used throughout this work. For a positive integer $K$, we write $[K]$ to denote $\{1, \ldots, K\}$ and $[\![K]\!]$ to denote $\{-K, \ldots, 0, \ldots, K\}$. For $a > 0$ denote by $\lfloor a \rfloor$ largest non-negative integer that is smaller or equal to $a$. For a univariate probability measure $\mu$, we denote by $\text{supp}(\mu)$ its support. For every $\beta > 0$, $m \in \mathbb{N}$, and $\boldsymbol{w} = (w_1, \ldots, w_m)^\top \in \mathbb{R}^m$, we denote by $\text{LSE}_\beta : \mathbb{R}^m \to \mathbb{R}$ the log-sum-exp function, defined as

$$\text{LSE}_\beta(\boldsymbol{w}) = \beta^{-1} \log \Big( \sum_{j=1}^m \exp(\beta w_j) \Big).$$

For every $m \in \mathbb{N}$, $\boldsymbol{w} = (w_1, \ldots, w_m)^\top \in \mathbb{R}^m$, we denote by $\boldsymbol{\sigma} = (\sigma_1, \ldots, \sigma_m) : \mathbb{R}^m \to \mathbb{R}^m$ the soft-argmax as $\sigma_j(\boldsymbol{w}) = \exp(w_j)/(\sum_{i=1}^m \exp(w_i))$. For any matrix $\mathbf{A}$, the notation $\mathbf{A} \geqslant 0$ means that $\mathbf{A}$ is positive coordinate-wise. For any $a \in \mathbb{R}$ and $\boldsymbol{w} \in \mathbb{R}^m$ we set $(a)_+ = \max\{0, a\}$ and $(\boldsymbol{w})_+ = ((w_1)_+, \ldots, (w_m)_+)^\top$. The notation $\widetilde{\mathcal{O}}$ hides (unimportant) constants and polylogarithmic factors. For a pair of random elements $(A, B)$, we denote by $\text{Law}(A)$, the law of $A$, by $\text{Law}(A \mid B)$, the conditional law of $A$ given $B$, and we write $A \perp\!\!\!\perp B$ to denote that variables $A$ and $B$ are independent. For two vectors $\boldsymbol{w}, \boldsymbol{w}' \in \mathbb{R}^m$, we write $\boldsymbol{w}/\boldsymbol{w}' = (w_j/w_j')_{j \in [m]} \in \mathbb{R}^m$ to denote element-wise division. The Euclidean norm of a vector and the Frobenius norm of a matrix are denoted by $\| \cdot \|$, while the spectral norm of a matrix is denoted by $\| \cdot \|_{\text{op}}$. We denote by $\mathcal{B}(\mathbb{R})$, the Borel sigma-algebra on $\mathbb{R}$, induced by the usual topology. We write $\log$ to denote the natural logarithm and $\log_a$, the base $a > 0$ logarithm.

## 2 Problem setup

Let $(\boldsymbol{X}, S, Y)$ be a triplet of nominally non-sensitive, nominally sensitive, and output characteristics, taking values in $\mathbb{R}^d \times [K] \times \mathbb{R}$ for some $K \geqslant 2$. We assume that $(\boldsymbol{X}, S, Y) \sim \mathbb{P}$, for some unknown distribution $\mathbb{P}$. The main quantities of interest are the following: the *regression function* $\eta(\boldsymbol{x}) \stackrel{\text{def}}{=} \mathbb{E}[Y \mid \boldsymbol{X} = \boldsymbol{x}]$; the marginal distribution of sensitive vectors $\boldsymbol{p} \stackrel{\text{def}}{=} (p_s)_{s \in [K]}$ with $p_s \stackrel{\text{def}}{=} \mathbb{P}(S = s)$; the conditional distribution of $S$ given $\boldsymbol{X}$, defined as $\boldsymbol{\tau}(\boldsymbol{x}) \stackrel{\text{def}}{=} (\tau_s(\boldsymbol{x}))_{s \in [K]}$ with $\tau_s(\boldsymbol{x}) \stackrel{\text{def}}{=} \mathbb{P}(S = s \mid \boldsymbol{X} = \boldsymbol{x})$. A *randomized* prediction function is a map $\pi : \mathcal{B}(\mathbb{R}) \times \mathbb{R}^d \to [0, 1]$ such that the map $B \mapsto \pi(B \mid \boldsymbol{x})$ for $B \in \mathcal{B}(\mathbb{R})$ is a probability measure on $(\mathbb{R}, \mathcal{B}(\mathbb{R}))$ for all $\boldsymbol{x} \in \mathbb{R}^d$. For any prediction $\pi$ we define a random variable $\widehat{Y}_\pi$ as

$$\text{Law}\left(\widehat{Y}_\pi \mid \boldsymbol{X} = \boldsymbol{x}, S = s\right) = \pi(\cdot \mid \boldsymbol{x}) \quad \boldsymbol{x} \in \mathbb{R}^d, s \in [K].$$

**Remark 2.1.** *Note that if $\pi(\cdot \mid \boldsymbol{x})$ is a Dirac measure for all $\boldsymbol{x} \in \mathbb{R}^d$, the above condition just means that $\widehat{Y}_\pi = g(\boldsymbol{X})$ almost surely for some deterministic $g : \mathbb{R}^d \to \mathbb{R}$. The above condition is not to be confused with the fairness constraint, which is not formulated point-wise. It is only viewed as an extension of the* unawareness *framework to the case of randomized predictions. The above condition completely specifies the distribution of the triplet $(\boldsymbol{X}, S, \widehat{Y}_\pi)$ but leaves the relation between $\widehat{Y}_\pi$ and $Y$ ambiguous. To be more formal, one needs to add the condition $(\widehat{Y}_\pi \perp\!\!\!\perp Y) \mid (\boldsymbol{X}, S)$, that is, the prediction $\widehat{Y}_\pi$ is independent from the true label $Y$, conditionally on $(\boldsymbol{X}, S)$. That would define a complete joint distribution of $(\boldsymbol{X}, S, Y, \widehat{Y}_\pi) \sim \mathbb{P}_\pi = \mathbb{P}_{(\boldsymbol{X}, S)} \otimes \mathbb{P}_{Y|(\boldsymbol{X}, S)} \otimes \pi(\cdot \mid \boldsymbol{X})$.*

We consider the following risk of a prediction function $\pi$

$$\mathcal{R}(\pi) \stackrel{\text{def}}{=} \mathbb{E}[(\widehat{Y}_\pi - \eta(\boldsymbol{X}))^2] = \mathbb{E}\left[\int_{\mathbb{R}} (\widehat{y} - \eta(\boldsymbol{X}))^2 \pi(\mathrm{d}\widehat{y} \mid \boldsymbol{X})\right].$$

A prediction function $\pi$ is said to satisfy the *demographic parity constraint*, if $\widehat{Y}_\pi \perp\!\!\!\perp S$.

That is, $\widehat{Y}_\pi$ is stochastically *independent* of $S$ viewed from the perspective of the joint distribution of $(\boldsymbol{X}, S, \widehat{Y}_\pi)$. On the high-level, the goal in this setup is to find a prediction function $\pi$, whose risk is small and whose violation of the demographic parity constraint is controlled as quantified by some measure of unfairness. The above problem is well understood in the case of *awareness*—the situation when $\pi$ is expressed as $\pi(\cdot \mid \boldsymbol{x}, s)$ (Chiappa et al., 2020, Chzhen et al., 2020a, 2021, Jiang et al., 2020, Le Gouic et al., 2020)—revealing an intimate connection of this problem with Wasserstein barycenters. Yet, when the sensitive attribute is not an input of the prediction function, the situation is drastically different. Some attempts have been made to either (so far only partially) characterise the optimal prediction function (Chzhen and Schreuder, 2020a, Gaucher et al., 2023, Zhao, 2021) or to design efficient algorithms for this problem (Agarwal et al., 2019, Maheshwari and Perrot, 2022, Narasimhan et al., 2020) that are only partially supported by a sound theory. One of the principal goals of this work is to design a computationally efficient algorithm that admits a (near) end-to-end theoretical guarantees. The main difficulty of the problem lies in very different natures of the risk and the fairness constraint—the latter involves image measures, while the former is a simple linear functional of $\pi$. In the case of awareness this issue can be bypassed by lifting the problem in the space of measures, working there directly and, then, returning to the initial space of prediction functions. Crucially, this is achieved only thanks to the fact that $S$ is known at inference time, which is not the case for the considered problem.

**Remark 2.2.** *In what follows we will exclusively focus on the squared risk and the regression setup. However, one can observe that the proposed methodology can be extended or even simplified for $\mathcal{R}(\pi) = \mathbb{E}[r(\boldsymbol{X}, \widehat{Y}_\pi)]$ and multi-class classification respectively under the demographic parity constraint. Here $r(\boldsymbol{x}, \widehat{y})$ quantifies fit of $\widehat{y}$ for an individual $\boldsymbol{x}$ and can be either known or unknown.*

## 3 Our methodology

The starting point of our work is similar to the one of Chzhen et al. (2020b) and relies on a simple observation—if $|\operatorname{supp}(\pi(\cdot \mid \boldsymbol{x}))| < \infty$ and stays the same for all $\boldsymbol{x}$, the independence constraint

is reduced to a finite amount of constraints that only involve the image of $\pi(\cdot \mid \boldsymbol{x})$. In particular, assuming that $\operatorname{supp}(\pi(\cdot \mid \boldsymbol{x})) = \widehat{\mathcal{Y}} \subset \mathbb{R}$ for all $\boldsymbol{x} \in \mathbb{R}^d$, $\widehat{Y}_\pi$ is independent from $S$ iff $\mathbb{P}(\widehat{Y}_\pi = \widehat{y} \mid S = s) = \mathbb{P}(\widehat{Y}_\pi = \widehat{y})$ for all $s \in [K]$ and all $\widehat{y} \in \widehat{\mathcal{Y}}$. In view of the definition of $\widehat{Y}_\pi$, the latter is equivalent to

$$\mathbb{E}[\pi(\widehat{y} \mid \boldsymbol{X}) \mid S] = \mathbb{E}[\pi(\widehat{y} \mid \boldsymbol{X})] \qquad s \in [K], \widehat{y} \in \widehat{\mathcal{Y}}, \tag{1}$$

which, assuming that $\widehat{\mathcal{Y}}$ is fixed, correspond to linear constraints on $\pi$. Combined with the observation that $\pi \mapsto \mathcal{R}(\pi)$ is also linear, we end up with a problem that is significantly easier to handle. Furthermore, again assuming that $\widehat{\mathcal{Y}}$ is fixed, the sketched direction gives a natural way to introduce some slack to the independence constraint—simply requiring an approximate equality in (1). Set

$$\mathcal{U}_s(\pi, \widehat{y}) \overset{\text{def}}{=} |\mathbb{E}\left[\pi(\widehat{y} \mid \boldsymbol{X}) \mid S = s\right] - \mathbb{E}\left[\pi(\widehat{y} \mid \boldsymbol{X})\right]|, \tag{2}$$

for all $s \in [K]$ and $\widehat{y} \in \widehat{\mathcal{Y}}$. Thus, for a fixed support (whose choice will be discussed in the next paragraph) and a fixed vector $\boldsymbol{\varepsilon} \overset{\text{def}}{=} (\varepsilon_1, \dots, \varepsilon_K)^\top$, our goal is to build an estimator of a solution to

$$\min_{\pi: \mathcal{B}(\mathbb{R}) \times \mathbb{R}^d \to [0,1]} \left\{ \mathcal{R}(\pi) : \operatorname{supp}(\pi(\cdot \mid \boldsymbol{x})) = \widehat{\mathcal{Y}} \text{ for } \boldsymbol{x} \in \mathbb{R}^d, \quad \mathcal{U}_s(\pi, \widehat{y}) \leqslant \varepsilon_s \text{ for } \widehat{y} \in \widehat{\mathcal{Y}}, s \in [K] \right\}. \tag{3}$$

Let us now describe the methodology for selecting $\widehat{\mathcal{Y}}$ and the trade-offs that are introduced.

**Introducing discretization.** Having in mind the above discussion, for every integer $L \geqslant 0$ and real $B > 0$, we introduce a uniform grid $\widehat{\mathcal{Y}}_L \overset{\text{def}}{=} B \cdot [\![L]\!]/L$ on $[-B, B]$, so that $|\widehat{\mathcal{Y}}_L| = 2L + 1$, which is viewed as a support of prediction functions $\pi(\cdot \mid \boldsymbol{x})$. For the sake of simplicity, we will assume that the regression function $\eta(\cdot)$ is bounded in $[-B, B]$ for some known $B > 0$.

**Assumption 3.1** (Bounded signal). *There exists $B > 0$ such that $|\eta(\boldsymbol{X})| \leqslant B$ almost surely.*

Thus, for a given $B$, the main parameter to tune is $L \geqslant 1$—the higher the $L$ is, the more accurate prediction functions can be produced, while lower values of $L$ ensure that the demographic parity requirement reduces to a small number of constraints. Thus, there is a trade-off that is introduced by $L$. A natural attempt to tackle the problem of fairness in this context would be to estimate a solution to (3) with $\widehat{\mathcal{Y}} = \widehat{\mathcal{Y}}_L$. Of course, $L$ needs to be chosen so that the aforementioned solution attains the risk that is close to the risk of some benchmark prediction function that does not involve any discretization. This will be discussed later in the text. For now, let us address another subtle issue. Even assuming a complete knowledge of the underlying distribution $\mathbb{P}$, solving (3) requires solving a linear program in dimension $\Omega(LK)$ which can be infeasible in practice for large values of $L$ and $K$. Instead of (3), we rather focus on the entropic regularized version of it. For $\beta > 0$, we consider

$$\min_{\pi: \mathcal{B}(\mathbb{R}) \times \mathbb{R}^d \to [0,1]} \left\{ \mathcal{R}_\beta(\pi) : \operatorname{supp}(\pi(\cdot \mid \boldsymbol{x})) = \widehat{\mathcal{Y}} \text{ for } \boldsymbol{x} \in \mathbb{R}^d, \quad \mathcal{U}_s(\pi, \widehat{y}) \leqslant \varepsilon_s \text{ for } \widehat{y} \in \widehat{\mathcal{Y}}, s \in [K] \right\}, \tag{4}$$

where $\mathcal{R}_\beta(\pi) = \mathcal{R}(\pi) + \frac{1}{\beta}\mathbb{E}[\Psi(\pi(\cdot \mid \boldsymbol{X}))]$ and for any discrete univariate distribution $\mu$, we define its negative entropy $\Psi(\mu) \overset{\text{def}}{=} \sum_{\widehat{y} \in \operatorname{supp}(\mu)} \mu(\widehat{y}) \log(\mu(\widehat{y}))$.

**Remark 3.1** (On abuse of notation). *Note that for every $\widehat{y} \in \widehat{\mathcal{Y}}_L$ there is a unique $\ell \in [\![L]\!]$ such that $\widehat{y} = \ell B/L$ and we will write $\pi(\ell \mid \boldsymbol{x})$ instead of $\pi(\widehat{y} \mid \boldsymbol{x})$. Similarly, we write $\mathcal{U}_s(\pi, \ell)$ instead of $\mathcal{U}_s(\pi, \widehat{y})$, defined in (2), when no confusion is possible and the support $\widehat{\mathcal{Y}}_L$ is fixed.*

An extremely attractive feature of the problem in (4) is the fact that the solution to it can be written explicitly as a function of optimal dual variables, with the latter being a solution of a stochastic convex program with Lipschitz gradient—the main observation of our approach, that shares many similarities with the smoothing technique of Nesterov (2005). This is summarized in the following lemma.

**Lemma 3.1.** *Let $L \in \mathbb{N}$ and $\beta > 0$. Let $\boldsymbol{\Lambda}^\star = (\lambda_{\ell s}^\star)_{\ell \in [\![L]\!], s \in [K]}$ and $\mathbf{V}^\star = (\nu_{\ell s}^\star)_{\ell \in [\![L]\!], s \in [K]}$ be two matrices that are solutions to*

$$\min_{\boldsymbol{\Lambda}, \mathbf{V} \geqslant 0} \left\{ F(\boldsymbol{\Lambda}, \mathbf{V}) \overset{\text{def}}{=} \mathbb{E}\left[\operatorname{LSE}_\beta\left(\left(\langle \boldsymbol{\lambda}_\ell - \boldsymbol{\nu}_\ell, \boldsymbol{t}(\boldsymbol{X})\rangle - r_\ell(\boldsymbol{X})\right)_{\ell \in [\![L]\!]}\right)\right] + \sum_{\ell \in [\![L]\!]} \langle \boldsymbol{\lambda}_\ell + \boldsymbol{\nu}_\ell, \boldsymbol{\varepsilon}\rangle \right\}, \tag{5}$$

*where $\boldsymbol{t}(\boldsymbol{x}) \overset{\text{def}}{=} 1 - \frac{\tau(\boldsymbol{x})}{\boldsymbol{p}}$, $r_\ell(\boldsymbol{x}) \overset{\text{def}}{=} \left(\eta(\boldsymbol{x}) - \frac{\ell B}{L}\right)^2$, and $\boldsymbol{\lambda}_\ell = (\lambda_{\ell s})_{s \in [K]}$, $\boldsymbol{\nu}_\ell = (\nu_{\ell s})_{s \in [K]}$. Then, (4) admits a solution in the form*

$$\pi_{\boldsymbol{\Lambda}^\star, \mathbf{V}^\star}(\ell \mid \boldsymbol{x}) \overset{\text{def}}{=} \sigma_\ell\left(\beta\left(\langle \boldsymbol{\lambda}_{\ell'}^\star - \boldsymbol{\nu}_{\ell'}^\star, \boldsymbol{t}(\boldsymbol{x})\rangle - r_{\ell'}(\boldsymbol{x})\right)_{\ell' \in [\![L]\!]}\right) \text{ for } \ell \in [\![L]\!]. \tag{6}$$

Assuming perfect knowledge of $\eta$ and $\tau$, the above lemma suggests a natural approach to estimating the $\pi_{\boldsymbol{\Lambda}^\star, \mathbf{V}^\star}$—we can run a (version of) stochastic gradient descent on $F(\cdot, \cdot)$ and then plug-in the resulting dual variables in the formula for $\pi_{\boldsymbol{\Lambda}^\star, \mathbf{V}^\star}$. Notably, a stochastic gradient of $F(\cdot, \cdot)$ can be obtained by simply sampling one $\boldsymbol{X}$ from $\mathbb{P}_{\boldsymbol{X}}$—it does not require labels for this step. Yet, even in the above idealized case, it is not clear which optimization criteria would allow us to prove that the resulting solution would yield good properties in terms of risk and fairness. As we will see, despite the problem in (5) being convex with Lipschitz gradient, it is crucial to control the norm of the gradient of $F$ for good statisitcal properties of the algorithm. That goes without saying that this relaxation has its price—the smaller the regularization parameter $\beta$ the less accurate the resulting solution, but the resulting dual optimization problem is easier and vice-versa.

**Properties of $F$ and $\pi_{\boldsymbol{\Lambda}^\star, \mathbf{V}^\star}$.** Let us summarized key properties of the objects introduced in Lemma 3.1. The first two results concern the population properties of $\pi_{\boldsymbol{\Lambda}^\star, \mathbf{V}^\star}$:

**Lemma 3.2** (Fairness quantification). *Let $L \in \mathbb{N}$, $\boldsymbol{\varepsilon} = (\varepsilon_s)_{s \in [K]} \in [0,1]^K, \beta > 0$, and $\pi_{\boldsymbol{\Lambda}^\star, \mathbf{V}^\star}$ be defined in Lemma 3.1. Then, $\mathcal{U}_s(\pi_{\boldsymbol{\Lambda}^\star, \mathbf{V}^\star}, \ell) \leqslant \varepsilon_s$ for all $s \in [K], \ell \in [\![L]\!]$.*

In words, the optimal entropic-regularized prediction function is feasible for (3), that is, it satisfies the relaxed fairness constraints as quantified by (2). Furthermore, we can show that its risk is also controlled by the regularization parameter $\beta > 0$.

**Lemma 3.3** (Risk gain). *Let $L \in \mathbb{N}, \beta > 0$, and $\pi_{\boldsymbol{\Lambda}^\star, \mathbf{V}^\star}$ be defined in Lemma 3.1. For any $\pi : \mathcal{B}(\mathbb{R}) \times \mathbb{R}^d \to [0,1]$ that is feasible for (3), we have*

$$\mathcal{R}(\pi_{\boldsymbol{\Lambda}^\star, \mathbf{V}^\star}) \leqslant \mathcal{R}(\pi) + \frac{\log|\widehat{\mathcal{Y}}_L|}{\beta} \,.$$

The above result is rather instructive, it quantifies the price of the introduced regularization. Intuitively, one wants to set $\beta$ high enough, so that the additive term in the above bound is vanishing. Unfortunately, we cannot set it arbitrarily high, since it will introduce instabilities from the optimization perspective—the function $F$ becomes less regular as $\beta$ growth. This is summarized below.

**Lemma 3.4** (Regularity of $F$). *Let $\sigma^2 \overset{\text{def}}{=} 2 \sum_{s \in [K]} \frac{1-p_s}{p_s}$. The objective function in (5) is convex and its gradient is $(\beta \sigma^2)$-Lipschitz.*

As mentioned, we see that the larger the $\beta$ is, the less regular the function $F$ is, making it harder to minimize. Thus, $\beta \geqslant 0$ controls the trade-off between the optimization error and statistical bias.

**Gradient of $F$ is crucial.** Let us show that the control of the gradient of $F$ is the most important and non-trivial part that allows to demonstrate strong statistical properties of the plug-in rule derived from the above strategy.

To this end, let us introduce parametric family of prediction functions, defined for any $\boldsymbol{\Lambda}, \mathbf{V} \geqslant 0$ as

$$\pi_{\boldsymbol{\Lambda}, \mathbf{V}}(\ell \mid \boldsymbol{x}) \overset{\text{def}}{=} \sigma_\ell \left( \beta \left( \langle \boldsymbol{\lambda}_{\ell'} - \boldsymbol{\nu}_{\ell'}, \boldsymbol{t}(\boldsymbol{x}) \rangle - r_{\ell'}(\boldsymbol{x}) \right)_{\ell' \in [\![L]\!]} \right) \text{ for } \ell \in [\![L]\!] \,. \tag{7}$$

We want to show that if $\boldsymbol{\Lambda}, \mathbf{V} \geqslant 0$ is nearly stationary point of $F$, then $\pi_{\boldsymbol{\Lambda}, \mathbf{V}}$ is nearly optimal in terms of risk and its violation of the demographic parity constraint is controlled. Note that the optimization problem in (5) is constrained, thus, unless the minimum lies in the interior of the domain, we cannot hope for the gradient of $F$ to go to zero. Instead, we introduce gradient mapping—a quantity that shares many properties of the gradient in the case of constraint optimization problem. For $\alpha > 0$,

$$\boldsymbol{G}_\alpha(\boldsymbol{\Lambda}, \mathbf{V}) \overset{\text{def}}{=} \frac{(\boldsymbol{\Lambda}, \mathbf{V}) - ((\boldsymbol{\Lambda}, \mathbf{V}) - \alpha \nabla F(\boldsymbol{\Lambda}, \mathbf{V}))_+}{\alpha} \,. \tag{8}$$

Our main observation is summarized in the next lemma.

**Lemma 3.5.** *Let $\sigma^2 \overset{\text{def}}{=} 2 \sum_{s \in [K]} \frac{1-p_s}{p_s}$, $L \in \mathbb{N}, \boldsymbol{\Lambda}, \mathbf{V} \geqslant 0$, then for any $\alpha > 0, \beta > 0$, the unfairness of $\pi_{\boldsymbol{\Lambda}, \mathbf{V}}$ satisfies*

$$\sum_{\ell \in [\![L]\!] s \in [K]} \left( \mathcal{U}_s(\pi_{\boldsymbol{\Lambda}, \mathbf{V}}, \ell) - \varepsilon_s \right)_+^2 \leqslant \|\boldsymbol{G}_\alpha(\boldsymbol{\Lambda}, \mathbf{V})\|^2 \,.$$

*Furthermore,*

$$\mathcal{R}(\pi_{\boldsymbol{\Lambda}, \mathbf{V}}) \leqslant \mathcal{R}(\pi_{\boldsymbol{\Lambda}^\star, \mathbf{V}^\star}) + \left( \|(\boldsymbol{\Lambda}, \mathbf{V})\| + \alpha \left\{ \sigma + \|\boldsymbol{\varepsilon}\| \sqrt{2|\widehat{\mathcal{Y}}_L|} \right\} \right) \|\boldsymbol{G}_\alpha(\boldsymbol{\Lambda}, \mathbf{V})\| + \frac{\log|\widehat{\mathcal{Y}}_L|}{\beta} \,.$$

Lemma 3.5 is very instructive on its own—we can obtain a good estimator of $\pi_{\mathbf{\Lambda}^\star, \mathbf{V}^\star}$ in terms of risk and unfairness by performing stochastic optimization on $F$ and controlling the norm of gradient mapping for a suitable parameter $\alpha > 0$. The final choice of the parameter $\alpha$ will depend on the optimization algorithm used and will be purely theoretical. In particular, for our purposes, it is sufficient to guarantee an existence of some value of $\alpha > 0$ that yields desired statistical properties. A naive approach in doing so relies on a well-known relation between $F(\mathbf{\Lambda}, \mathbf{V}) - F(\mathbf{\Lambda}^\star, \mathbf{V}^\star)$ and $\|\boldsymbol{G}_\alpha(\mathbf{\Lambda}, \mathbf{V})\|^2$ using the Lipshitzness of the gradient of $F$ (see e.g., Beck, 2014, Lemma 9.11). More concretely, forgetting about the constraints[1], one has

$$\|\nabla F(\mathbf{\Lambda}, \mathbf{V})\|^2 \leqslant 2M\big(F(\mathbf{\Lambda}, \mathbf{V}) - F(\mathbf{\Lambda}^\star, \mathbf{V}^\star)\big), \tag{9}$$

where $M$ is the Lipschitz constant of $\nabla F$. Thus, the above inequality suggests that it is sufficient to control the standard optimization error in order to control the norm of the gradient. Unfortunately this approach is deemed to fail for two reasons: the first being that we control only the squared norm of the gradient map and not the norm itself, thus loosing in the rate of convergence; the second, and more subtle reason, is the separation of the purely "statistical" rate that depends only on the variance of the stochastic gradient and scales as $1/\sqrt{T}$, with $T$ being the number of future samples from $\mathbb{P}_{\boldsymbol{X}}$, and "optimization" rate of convergence that depends on $M$ and the diameter of the problem and typically scales as $1/T$ or even $1/T^2$ if acceleration is used.

Indeed, in our setup, Lipschitz constant $M$ of $\nabla F$ is not a fixed constant, but a parameter to be set—it relates to $\beta$ (cf. Lemma 3.4). Ideally, seeing Lemma 3.3, we want to set $\beta = \Theta(\sqrt{T})$, leading to $M = \Omega(\sqrt{T})$. Thus, in view of (9), a term of the form $M/\sqrt{T}$ appears in the convergence rate, which destroys consistency of the resulting estimator. Arguably, this is less of an issue in case of convex optimization with constant Lipschitz constant $M$, especially if we only want the norm to go to zero. This discussion highlights that it is crucial to keep the separation between the statistical part of the rate and the optimization part of the rate, while controlling the norm of the gradient. Lucky for us, it is known that for convex problems one can indeed control the gradient mapping keeping this separation of the rate (Allen-Zhu, 2021, Foster et al., 2019). Note that it is not the case for non-convex problems as demonstrated by Arjevani et al. (2023).

**Summary of our approach and why is it different from others.** Now, keeping in mind the above, rather long justification, we are in position to sketch our approach and the formal presentation is deferred to the next section. For well selected parameters $\beta > 0$, $L \in \mathbb{N}$, we are going to perform stochastic optimization of $F$, relying on the SGD3 algorithm of Allen-Zhu (2021). In order to compute the stochastic gradient of $F$, we are simply going to sample one $\mathbb{P}_{\boldsymbol{X}}$ and it appears that this stochastic gradient has a well-behaved variance (see Appendix B-C for details). To make our approach completely data-driven (or at least to understand the order of magnitude of the parameters), we will compute or bound all the oracle quantities that appear in the used optimization algorithm (essentially related to the step-size tuning). We will show that for any sufficiently small $\alpha > 0$, the term $\mathbb{E}\|\boldsymbol{G}_\alpha(\widehat{\mathbf{\Lambda}}, \widehat{\mathbf{V}})\|^2$ is controlled and then rely on Lemma 3.5 and some additional results to demonstrate that the resulting $\pi_{\widehat{\mathbf{\Lambda}}, \widehat{\mathbf{V}}}$ possesses good statistical properties.

**Remark 3.2** (On the dynamic of algorithm). *Note that for $\mathbf{\Lambda} = \mathbf{V} = \mathbf{0}$, the corresponding*

$$\big(\pi_{\mathbf{0}, \mathbf{0}}(\ell \mid \boldsymbol{x})\big)_{\ell \in [\![L]\!]} = \boldsymbol{\sigma}\left(\beta\left(-(\eta(\boldsymbol{x}) - \ell' B/L)^2\right)_{\ell' \in [\![L]\!]}\right).$$

*That is, the above prediction puts the most amount of mass on the atom $\ell$ which minimizes $(\eta(\boldsymbol{x}) - \ell B/L)^2$—the most accurate, but unfair prediction. Since our algorithm is based on a SGD-type algorithm, initialized at $\mathbf{\Lambda}_0 = \mathbf{V}_0 = \mathbf{0}$, then we expect that during the dynamic of the algorithm, the risk of $\pi_{\mathbf{\Lambda}_t, \mathbf{V}_t}$ increases, while the unfairness decreases. This phenomena coincides with the intuition of post-processing—we want to gain in fairness, while sacrificing some accuracy.*

As it has been already mentioned, the idea of discretizing the image of (randomized) predictions is not novel and has been successfully deployed by Agarwal et al. (2019) for an in-processing estimator and by Chzhen et al. (2020b) for a post-processing estimator. We use this insight as a building block, but significantly deviate from both algorithms. Compared to Agarwal et al. (2019), our algorithm is positioned in the realm of post-processing and even *online* post-processing, where i.i.d. samples

---

[1]Constraints introduce additional challenges, but are not relevant for this discussion.

from $\mathbb{P}_{\boldsymbol{X}}$ comes in a stream and we do not need to store them in memory. Also, while their algorithm is partially inspired by theory, the same theory suggests that this algorithm is not computationally efficient and it relies on some black-box parts that assume perfect solutions to some optimization problems. That being said, the algorithm of Agarwal et al. (2019) seem to be the gold standard method for the generic in-processing method in this problem. Compared to Chzhen et al. (2020b), we have made a sequence of improvements. First, our setup is unawareness, which is not the case in their paper; second, our algorithm is able to handle multiple protected attributes as well as approximate fairness constraints; finally, and most importantly, we do not make black-box assumptions about having access to exact minimizers of convex problems and provide an end-to-end analysis of out approach. Let us also remark that our method cannot be considered as a simple extension of Chzhen et al. (2020b) as we rely on different phenomenons and provide a very different algorithm. On a more subjective note, we believe that our approach is a nice example of a real convex optimization problem, where the norm of the gradient plays the central role, while the optimization error in term of the objective function does not matter[2]. This is precisely the phenomena highlighted by Nesterov (2012).

## 4 Proposed algorithm

---
**Algorithm 1:** DP post-processing$(L, T, \beta, \boldsymbol{p}, B, \eta, \boldsymbol{\tau})$
---
1: **Input:** discretization parameter $L \geqslant 1$; regularization $\beta > 0$, number of stochastic gradient evaluations $T \geqslant 1$; marginal distribution $\boldsymbol{p}$ of $S$; regression function $\eta$; conditional distribution $\boldsymbol{\tau}$ of $S \mid \boldsymbol{X}$; bound $B > 0$ on $\eta$.
2: Build uniform grid $\widehat{\mathcal{Y}}_L$ over $[-B, B]$;
3: Set parameters: $\sigma^2 = 2 \sum_{s \in [K]} \frac{1-p_s}{p_s}$, $M = \beta \sigma^2$;
4: Set $(\boldsymbol{\Lambda}, \mathbf{V}) \mapsto F(\boldsymbol{\Lambda}, \mathbf{V})$ as defined in Lemma 3.1
5: Run a black-box optimizer $\mathcal{A}(F, \sigma^2, M, T)$ on function $F$ having access to $T$ stochastic gradient evaluations (see (11)) with variance $\sigma^2$ and smoothness parameter $M$ to obtain $(\widehat{\boldsymbol{\Lambda}}, \widehat{\mathbf{V}})$;
6: **return** $\pi_{(\widehat{\boldsymbol{\Lambda}}, \widehat{\mathbf{V}})}(\cdot \mid \cdot)$ as defined in (7);
---

In this section, we provide all the details about the proposed algorithm in case $\eta$ and $\boldsymbol{\tau}$ are known. If they are unknown, these quantities are replaced by their estimates $\widehat{\eta}$ and $\widehat{\boldsymbol{\tau}}$ that are constructed on a separate labeled dataset. First, for $\boldsymbol{\Lambda} = (\lambda_{\ell s})_{\ell \in [\![L]\!], s \in [K]}$, $\mathbf{V} = (\nu_{\ell s})_{\ell \in [\![L]\!], s \in [K]}$, let us provide the expression for the gradient of $F$:

$$\nabla_{\square_{\ell s}} F(\boldsymbol{\Lambda}, \mathbf{V}) = \triangle \mathbb{E} \left[ \sigma_\ell \left( \beta \left( \langle \boldsymbol{\lambda}_{\ell'} - \boldsymbol{\nu}_{\ell'}, \boldsymbol{t}(\boldsymbol{X}) \rangle - r_{\ell'}(\boldsymbol{X}) \right)_{\ell'=-L}^{L} \right) t_s(\boldsymbol{X}) \right] + \varepsilon_s, \tag{10}$$

where $\square \in \{\lambda, \nu\}$ and $\triangle = 1$ if $\square = \lambda$ and $\triangle = -1$ otherwise. Thus, a *stochastic gradient* $\boldsymbol{g}(\boldsymbol{\Lambda}, \mathbf{V}) = (g_{\lambda_{\ell s}}(\boldsymbol{\Lambda}, \mathbf{V}), g_{\nu_{\ell s}}(\boldsymbol{\Lambda}, \mathbf{V}))_{\ell \in [\![L]\!], s \in [K]}$ of $F$ at a point $(\boldsymbol{\Lambda}, \mathbf{V})$ can be computed by erasing expectation in (10), *i.e.*, by sampling one $\boldsymbol{X} \sim \mathbb{P}_{\boldsymbol{X}}$, using the same convention as above about $\square, \triangle$:

$$g_{\square_{\ell s}}(\boldsymbol{\Lambda}, \mathbf{V}) = \triangle \sigma_\ell \left( \beta \left( \langle \boldsymbol{\lambda}_{\ell'} - \boldsymbol{\nu}_{\ell'}, \boldsymbol{t}(\boldsymbol{X}) \rangle - r_{\ell'}(\boldsymbol{X}) \right)_{\ell'=-L}^{L} \right) t_s(\boldsymbol{X}) + \varepsilon_s. \tag{11}$$

The next result controls the variance of the above stochastic gradient.

**Lemma 4.1.** *Let* $\sigma^2 \stackrel{\text{def}}{=} 2 \sum_{s \in [K]} \frac{1-p_s}{p_s}$. *It holds that* $\mathbb{E}\|\boldsymbol{g}(\boldsymbol{\Lambda}, \mathbf{V}) - \nabla F(\boldsymbol{\Lambda}, \mathbf{V})\|^2 \leqslant \sigma^2$.

The proposed method is summarized in Algorithm 1. It uses a black-box stochastic optimization algorithm $\mathcal{A}$, that operates on a convex function $F$ and a stochastic first-order oracle. The stochastic-first order oracle is implemented by (11) and only requires to sample $\boldsymbol{X} \sim \mathbb{P}$ in an i.i.d. manner. We also pass two additional parameters to this algorithm: namely, we pass the variance $\sigma^2$ from Lemma 4.1 and the Lipschitz constant of the gradient of $F$ from Lemma 3.4. Then one can use any such algorithm. However, as shown in Lemma 3.5, those algorithms that are tailored to control expected norm of gradient mapping are preferred. For example, one can use SGD3 of Allen-Zhu (2021) or an improved version of Foster et al. (2019) that relies on restarted accelerated SGD of Ghadimi and Lan (2012).

---
[2]To be more precise, the optimization error is automatically handled by the control of the gradient.

# 5  Theoretical guarantees.

Let us first provide main results for Algorithm 1 assuming that $\eta$ and $\boldsymbol{\tau}$ are known. Note that Algorithm 1 can rely on any optimization algorithm. We provide a complete analysis using a refined version of SGD3 algorithm of Allen-Zhu (2021) that is due to Foster et al. (2019) with additional modifications taking into account the specific structure of our problem. We state the main result in existential form and postpone all the details on the implementation of the algorithm and a primer on optimization to the supplementary material (Appendix C-D).

**Theorem 5.1.** *Let $\varepsilon = (\varepsilon_s)_{s \in [K]} \in [0,1]^K$ and $\sigma^2 = 2 \sum_{s \in [K]} (1 - p_s)/p_s$. Setting $\beta = \frac{T}{8 \log_2(T)}$ and $L = \sqrt{T}$, there exists an optimizer $\mathcal{A}$ to be used in Algorithm 1 that, for $T$ larger than some absolute constant, ensures*

$$\mathbf{E}^{1/2}\left[ \sum_{\ell \in [\![L]\!] s \in [K]} \left( \mathcal{U}_s(\pi_{\mathbf{\Lambda},\mathbf{V}}, \ell) - \varepsilon_s \right)_+^2 \right] \leqslant \widetilde{\mathcal{O}}\left( \frac{\sigma}{\sqrt{T}} \left( 1 + \frac{\sigma}{\sqrt{T}} \|(\mathbf{\Lambda}^\star, \mathbf{V}^\star)\| \right) \right).$$

*Furthermore, if Assumption 3.1 is satisfied and let*

$$\mathcal{R}^\star \stackrel{\text{def}}{=} \inf_{h: \mathbb{R}^d \to [-B,B]} \left\{ \mathcal{R}(h) \; : \; \sup_{t \in \mathbb{R}} |\mathbb{P}(h(\mathbf{X}) \leqslant t \mid S = s) - \mathbb{P}(h(\mathbf{X}) \leqslant t)| \leqslant \frac{\varepsilon_s}{2}, \quad \forall s \in [K] \right\} \tag{12}$$

*and $\mathcal{E}(\pi_{\widehat{\mathbf{\Lambda}},\widehat{\mathbf{V}}}) \stackrel{\text{def}}{=} \mathbb{E}\left[ \mathcal{R}(\pi_{\widehat{\mathbf{\Lambda}},\widehat{\mathbf{V}}}) \right] - \mathcal{R}^\star$, then for the same algorithm*

$$\mathcal{E}(\pi_{\widehat{\mathbf{\Lambda}},\widehat{\mathbf{V}}}) \leqslant \widetilde{\mathcal{O}}\left( \left( \frac{\sigma}{\sqrt{T}} \mathbf{E}^{1/2}\left[ \|(\widehat{\mathbf{\Lambda}}, \widehat{\mathbf{V}})\|^2 \right] + \frac{\|\varepsilon\|}{T^{5/4}} \right) \left( 1 + \frac{\sigma}{\sqrt{T}} \|(\mathbf{\Lambda}^\star, \mathbf{V}^\star)\| \right) + \frac{B}{\sqrt{T}} \right).$$

Theorem 5.1 gives two results: the first one being on the unfairness of the proposed estimator and the second one on the risk of thereof compared to a benchmark prediction function in (12). The benchmark that we pick is rather natural, we compare to the risk of a deterministic prediction that minimizes the risk and whose unfairness is controlled by a Kolmogorov-Smirnov distance. One first main observation is that both fairness and risk decrease at the rate $1/\sqrt{T}$ and $T$ is the number of unlabeled data. From our numerical experiments, we observed that we can keep the number of unlabeled data unchanged and iterate several times through them. As a result, we increase artificially $T$—without generating new data—which gives a significant empirical improvement. We also remark that $\sigma$ is the parameter that depends on the number of groups. For example, in the case of uniform distribution of sensitive groups $\sigma = O(K)$. We finally remark that both bounds involve a single unknown quantity—$\|(\mathbf{\Lambda}^\star, \mathbf{V}^\star)\|$, which from standard duality argument can be shown to be bounded by $O(1/\min_{s \in [K]} \{\varepsilon_s\})$ (see e.g., Nedić and Ozdaglar, 2009, Lemma 3). Thus, having this norm multiplied by $T^{-1/2}$ is a very attractive property of the bound. It allows to set $\varepsilon \approx T^{-1/2}$ without damaging the parametric convergence rate.

To derive the above result, we slightly extend the analysis of Foster et al. (2019), who, relying on the SGD3 algorithm of Allen-Zhu (2021), gave an optimal algorithm that controls the expected norm of the gradient in the convex case. More concretely, we incorporate a projection step into their analysis and extend the control to the squared norm of the gradient map. Interestingly, due to our smoothing step and the choice of the parameter $\beta$, we noticed that there is no need to restart the accelerated SGD as it is done by Foster et al. (2019) because it leads to identical statistical convergence rates. The interested reader can take a closer look into the Appendix C, where all the optimization results are either recalled or derived for the sake of completeness. Finally, having a control of the squared norm of the gradient map, the proof of Theorem 5.1 follows from Lemma 3.5 and a careful and practical choice of all the parameters of the algorithm.

**Extension to unknown $\eta$ and $\boldsymbol{\tau}$.** In this part we show that if we replace $\eta$ and $\boldsymbol{\tau}$ with their estimates $\widehat{\eta}$ and $\widehat{\boldsymbol{\tau}}$ and run DP post-processing$(L, T, \beta, \boldsymbol{p}, B, \widehat{\eta}, \widehat{\boldsymbol{\tau}})$ algorithm with the same choice of parameters, Theorem 5.1 remains if we pay additional price for the estimation of $\eta$ and $\boldsymbol{\tau}$. From now on, we assume that $\widehat{\eta}$ and $\widehat{\boldsymbol{\tau}}$ are provided and are trained on its own labeled data sample, while the refitting is performed on an independent stream of i.i.d. data from $\mathbb{P}_{\mathbf{X}}$. So, we essentially treat $\widehat{\eta}$ and $\widehat{\boldsymbol{\tau}}$ as deterministic functions. Let us introduce a family of prediction functions

$$\widehat{\pi}_{\mathbf{\Lambda},\mathbf{V}}(\ell \mid \boldsymbol{x}) \stackrel{\text{def}}{=} \sigma_\ell\left( \beta \left( \left\langle \boldsymbol{\lambda}_{\ell'} - \boldsymbol{\nu}_{\ell'}, \widehat{\boldsymbol{t}}(\boldsymbol{x}) \right\rangle - \widehat{r}_{\ell'}(\boldsymbol{x}) \right)_{\ell' \in [\![L]\!]} \right) \quad \text{for } \ell \in [\![L]\!]. \tag{13}$$

Note that for fixed matrices $\mathbf{\Lambda}, \mathbf{V}$, the above prediction function is fully data-driven. With this plug-in strategy, our approach becomes fully data-driven and, in Appendix E, we show that the guarantees presented in the main body still hold paying additional price for estimation of $\eta$ and $\tau$. To be more precise, we consider the plug-in version of (5), defined as

$$\min_{\mathbf{\Lambda}, \mathbf{V} \geqslant 0} \left\{ \mathbb{E}_{\boldsymbol{X}} \left[ \mathrm{LSE}_\beta \left( \left( \left\langle \boldsymbol{\lambda}_\ell - \boldsymbol{\nu}_\ell, \widehat{\boldsymbol{t}}(\boldsymbol{X}) \right\rangle - \widehat{r}_\ell(\boldsymbol{X}) \right)_{\ell=-L}^{L} \right) \right] + \sum_{\ell=-L}^{L} \langle \boldsymbol{\lambda}_\ell + \boldsymbol{\nu}_\ell, \boldsymbol{\varepsilon} \rangle \right\} . \quad (\widehat{\mathcal{P}}_{LSE})$$

Let us denote by $\widehat{F}$, the objective function of the above problem. Thus, main interesting part is to demonstrate that a control of the gradient map of $\widehat{F}$, denoted by $\|\boldsymbol{G}_{\widehat{F},\alpha}(\widehat{\mathbf{\Lambda}}, \widehat{\mathbf{V}})\|$, gives a control of risk and unfairness of $\widehat{\pi}_{\widehat{\mathbf{\Lambda}}, \widehat{\mathbf{V}}}$, quantifying the price induced by the plug-in estimation. This is precisely the purpose of the following two results:

**Lemma 5.1.** *Let $L \in \mathbb{N}$, $\mathbf{\Lambda}, \mathbf{V} \geqslant 0$, then for any $\alpha > 0, \beta > 0$ it holds that*

$$\sqrt{\sum_{\ell \in [\![L]\!]} \sum_{s \in [K]} (\mathcal{U}_s(\widehat{\pi}_{\mathbf{\Lambda},\mathbf{V}}, \ell) - \varepsilon_s)_+^2} \leqslant \|\boldsymbol{G}_{\widehat{F},\alpha}(\mathbf{\Lambda}, \mathbf{V})\| + \mathbb{E}^{1/2}\|\widehat{\boldsymbol{t}}(\boldsymbol{X}) - \boldsymbol{t}(\boldsymbol{X})\|^2 .$$

**Lemma 5.2.** *Let $\widehat{\sigma}^2 = 2 \sum_{s \in [K]} \frac{\mathbb{E}_{\boldsymbol{X}} (p_s - \widehat{\tau}_s(\boldsymbol{X}))^2}{p_s^2}, L \in \mathbb{N}, \mathbf{\Lambda}, \mathbf{V} \geqslant 0$, then for any $\alpha > 0, \beta > 0$ it holds that*

$$\mathcal{R}(\widehat{\pi}_{\mathbf{\Lambda},\mathbf{V}}) \leqslant \mathcal{R}(\pi_{\mathbf{\Lambda}^\star,\mathbf{V}^\star}) + \left( \|(\mathbf{\Lambda}, \mathbf{V})\| + \alpha \left\{ \widehat{\sigma} + \|\boldsymbol{\varepsilon}\| \sqrt{2|\widehat{\mathcal{Y}_L}|} \right\} \right) \left\| \boldsymbol{G}_{\widehat{F},\alpha}(\mathbf{\Lambda}, \mathbf{V}) \right\| + \frac{\log |\widehat{\mathcal{Y}_L}|}{\beta}$$

$$+ 2\mathbb{E}_{\boldsymbol{X}} \left[ \max_{\ell \in [\![L]\!]} |r_\ell(\boldsymbol{X}) - \widehat{r}_\ell(\boldsymbol{X})| \right] + \sqrt{2}\|(\mathbf{\Lambda}, \mathbf{V})\| \cdot \mathbb{E}^{1/2}\|\boldsymbol{t}(\boldsymbol{X}) - \widehat{\boldsymbol{t}}(\boldsymbol{X})\|^2 .$$

Note that the two above results are extensions of Lemma 3.5, where both $\boldsymbol{t}$ and $\eta$ were assumed to be known. These results are following the spirit of post-processing bounds—the quality of the final approach depends on the initial estimator and the optimization algorithm used to post-process and the two errors are clearly separated. Proofs of both results with additional details and discussions is provided in the supplementary material.

## 6 Numerical illustration

In this section we conduct empirical study of the proposed algorithm, denoted by DP-postproc, and demonstrate its relevance in practical problems [3]. We have implemented both SGD3 of Allen-Zhu (2021) and an improved version by Foster et al. (2019), observing that the latter significantly outperforms the former. We also tested the approach that is suggested by the theory—SGD3 and accelerated SGD, without restart and it show nearly identical performance as the restarted version of Foster et al. (2019). Thus, for numerical evaluation, we stick to the latter.

We conduct our study on two datasets: *Law School* dataset (Wightman (1998)) and *Communities and Crime* dataset (Redmond (2009)). In the *Law School* dataset, the aim is to predict students' GPA on a scale of 0 to 4, normalized to $[0, 1]$, while in the *Communities and Crime* dataset, we focus on predicting the normalized number of violent crimes per population within the range of $[0, 1]$. In both datasets, ethnicity is a sensitive attribute, distinguishing between white and non-white individuals or communities (majority-wise).

Our pipeline is the following: First, we randomly split the data into training, unlabeled and testing sets with proportions of $0.4 \times 0.4 \times 0.2$. We use $\mathcal{D}_{\mathrm{train}} = \{(\boldsymbol{x}_i, s_i, y_i)_{i=1}^n\}$ to train a base (unfair) regressor to estimate $\eta$ and to train a classifier to estimate $\tau$. We use simple *LinearRegression* and *LogisticRegression* from *scikit-learn* for training the regressor and the classifier. Finally, we use the trained regressor and classifier to train the Algorithm 1 with $\mathcal{D}_{\mathrm{unlabeled}} = (\boldsymbol{x})_{i=n+1}^{n+T}$ for $N$ (note that our theory suggests that $N = T$ is enough, but we have noticed that larger $N$ can be more beneficial in practice) iterations. We use $\mathcal{D}_{\mathrm{test}} = \{(\boldsymbol{x}_i', s_i', y_i')_{i=1}^m\}$ to collect evaluation statistics.

In Figure 1 we illustrate the post-processing dynamics of our method. We have 2 plots for each test dataset: the history of risk $(\mathcal{R}(\widehat{\pi}))$ and of the unfairness $(U_0(\widehat{\pi})$ and $U_1(\widehat{\pi}))$ *w.r.t.* number of iterations. We illustrate the convergence for $\boldsymbol{\varepsilon} = (2^{-8}, 2^{-8})$ unfairness threshold. The explicit formulas of the evaluation measures are provided in Appendix G.

---

[3]The code is available at https://github.com/taturyan/unaware-fair-reg.

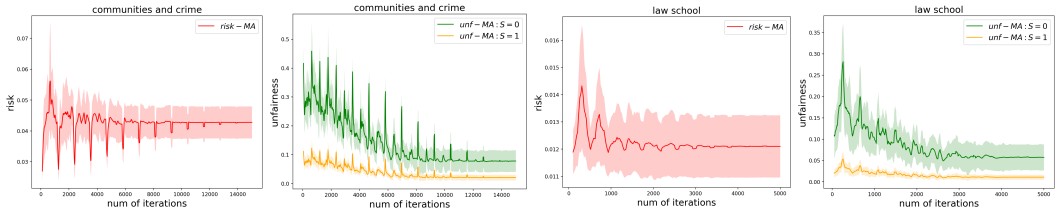

Figure 1: Risk and unfairness of our estimator on *Communities and Crime* and *Law School* datasets.

**Comparison with Agarwal et al. (2019).** Surprisingly, we were unable to find many open source competitors that target regression with demographic parity constraint in unaware setting, even the `FairLearn`—a popular `python` package—does not deal with the demographic parity constraint in regression. The only easily accessible algorithm that deals with our problem was kindly provided by Agarwal et al. (2019) (from now on referenced as ADW). We train ADW method in two ways: we use $\mathcal{D}_{\text{train}}$ and $\mathcal{D}_{\text{unlabeled}}$ as training set for ADW-1, whereas for ADW-2 we use only $\mathcal{D}_{\text{train}}$. The second situation is realistic, when unlabeled data is available and unlike ADW, our approach is able to take advantage of it. We take the set $\{(2^{-i}, 2^{-i})_{i \in \mathcal{I}}\}$, where $\mathcal{I} = \{1, 2, 4, 8, 16\}$ as unfairness thresholds for training both datasets. We train ADW-1 and ADW-2 for each pair of epsilons for 10 times. With our available computing power and the code provided by the authors, the algorithm runs for 13.5 hours (see Appendix G for additional details).[4]

On Figure 2 we illustrate the comparison of risk and unfairness between ADW-1, ADW-2, base (LinearRegression) and our model. We plot the mean and standard deviation of risk and unfairness for each epsilon threshold on both datasets. We observe that our method is competitive or eventually outperforms ADW in both training regimes.

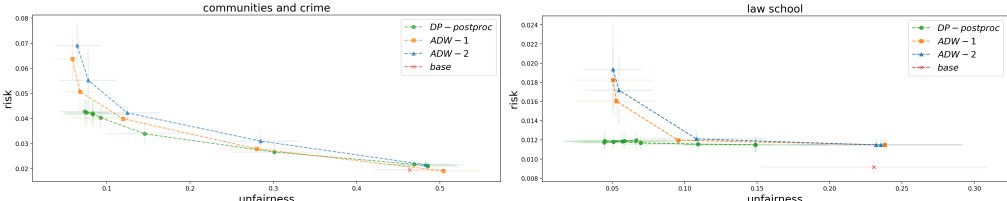

Figure 2: Comparison with ADW model on *Communitites and Crime* and *Law School* datasets.

## 7 Conclusion

Deriving a dual convex surrogate, we have provided a generic way to build a post-processing estimator of any off-the-shelf method that achieves the demographic parity constraint. Our approach is fully data and theory driven, revealing a key role of stationary point guarantees in stochastic convex optimization. Following Remark 2.2, we intend to extend our approach, which is general enough, to other learning problems, beyond algorithmic fairness.

**Limitations.** From the theoretical perspective, the knowledge of $B$ seems to be the main limitation. While it is available for many applications, it does not have to be the case all the time. Replacing this assumption with some tail conditions, could be more realistic. From the applied perspective, it would be beneficial to further investigate stationary point guarantees for convex optimization to yield a better practical performance.

**Acknowledgements** The work of Gayane Taturyan has been supported by the French government under the "France 2030" program, as part of the SystemX Technological Research Institute within the Confiance.ai project.

---

[4]The experiments are conducted on a Processor 11th Gen Intel(R) Core(TM) i7-1195G7 2.90GHz with 16GB RAM.

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

# A   Proofs for results in Section 3

First we explicit the first order optimality conditions for the problem in (5).

**Lemma A.1.** *Let* $(\mathbf{\Lambda}^\star, \mathbf{V}^\star) \geqslant 0$ *be any minimizer of* (5) *and* $\pi^\star = \pi_{\mathbf{\Lambda}^\star, \mathbf{V}^\star}$ *be defined in* (6). *Then, there exist* $\mathbf{\Gamma} = (\gamma_{\ell s})_{\ell \in \llbracket L \rrbracket, s \in [K]}, \mathbf{\Gamma}' = (\gamma'_{\ell s})_{\ell \in \llbracket L \rrbracket, s \in [K]}$—*element-wise non-negative matrices such that*

$$
\begin{cases}
\mathbb{E}_{\mathbf{X}}\left[\pi^\star(\ell \mid \mathbf{X})\mathbf{t}(\mathbf{X})\right] = -\boldsymbol{\varepsilon} + \boldsymbol{\gamma}_\ell \\
\mathbb{E}_{\mathbf{X}}\left[\pi^\star(\ell \mid \mathbf{X})\mathbf{t}(\mathbf{X})\right] = \boldsymbol{\varepsilon} - \boldsymbol{\gamma}'_\ell \\
\gamma_{\ell s}\lambda^\star_{\ell s} = 0 \\
\gamma'_{\ell s}\nu^\star_{\ell s} = 0
\end{cases}
\qquad \forall \ell \in \llbracket L \rrbracket, s \in [K],
\tag{14}
$$

*where* $\boldsymbol{\gamma}_\ell = (\gamma_{\ell s})_{s \in [K]}, \boldsymbol{\gamma}'_\ell = (\gamma'_{\ell s})_{s \in [K]}$.

*Proof.* We first observe that the optimization problem in (5) is convex and smooth. Thus, Karush–Kuhn–Tucker conditions are sufficient for optimally. Furthermore, since Slatter's condition is satisfied, the latter is also necessary, as the strong duality holds. In particular, there exist $\mathbf{\Gamma} = (\gamma_{\ell s})_{\ell \in \llbracket L \rrbracket, s \in [K]}, \mathbf{\Gamma}' = (\gamma'_{\ell s})_{\ell \in \llbracket L \rrbracket, s \in [K]}$—element-wise non-negative matrices such that

$$
\begin{cases}
\nabla_{\mathbf{\Lambda}} F(\mathbf{\Lambda}^\star, \mathbf{V}^\star) - \mathbf{\Gamma} = \mathbf{0} \\
\nabla_{\mathbf{V}} F(\mathbf{\Lambda}^\star, \mathbf{V}^\star) - \mathbf{\Gamma}' = \mathbf{0} \\
\mathbf{\Lambda}^\star, \mathbf{V}^\star \geqslant 0 \\
\gamma_{\ell s}\lambda^\star_{\ell s} = 0 \\
\gamma'_{\ell s}\nu^\star_{\ell s} = 0
\end{cases}
\qquad \forall \ell \in \llbracket L \rrbracket, s \in [L].
$$

To conclude, it is sufficient to evaluate the gradient on $F$, whose expression is given in (10) and use the definition of $\pi^\star$. ∎

***Proof of Lemma 3.1.*** To prove this result, we introduce the Lagrangian for the problem in (4).

$$
\mathcal{L}(\pi, \mathbf{\Lambda}, \mathbf{V}) = \mathcal{R}_\beta(\pi) + \mathbb{E}_{\mathbf{X}}\left[\sum_{\ell \in \llbracket L \rrbracket} \langle \boldsymbol{\nu}_\ell - \boldsymbol{\lambda}_\ell, \mathbf{t}(\mathbf{X})\rangle \pi(\ell \mid \mathbf{X})\right] - \sum_{\ell \in \llbracket L \rrbracket} \langle \boldsymbol{\lambda}_\ell + \boldsymbol{\nu}_\ell, \boldsymbol{\varepsilon}\rangle,
$$

where we used the fact, that using the definition of $\mathcal{U}_s$, for any randomized prediction function $\pi$, we can write

$$
\mathcal{U}_s(\pi, \ell) = \left| \frac{\mathbb{E}_{\mathbf{X}}\left[\pi(\ell \mid \mathbf{X})\mathbb{I}\{S = s\}\right]}{\mathbb{P}(S = s)} - \mathbb{E}_{\mathbf{X}}\left[\pi(\ell \mid \mathbf{X})\right] \right| = \left| \mathbb{E}_{\mathbf{X}}\left[\pi(\ell \mid \mathbf{X})t_s(\mathbf{X})\right] \right|.
\tag{15}
$$

Thus, denoting by $(\star)$ the value in (4), we have

$$
(\star) = \min_\pi \max_{\mathbf{\Lambda}, \mathbf{V} \geqslant 0} \mathcal{L}(\pi, \mathbf{\Lambda}, \mathbf{V}) = \max_{\mathbf{\Lambda}, \mathbf{V} \geqslant 0} \min_\pi \mathcal{L}(\pi, \mathbf{\Lambda}, \mathbf{V}),
$$

where the second equality holds thanks to Sion's minmax theorem. Let us solve the inner minimization problem on the right-hand-side. We can write

$$
\mathcal{L}(\pi, \mathbf{\Lambda}, \mathbf{V}) = \mathbb{E}_{\mathbf{X}}\left[\sum_{\ell \in \llbracket L \rrbracket} (r_\ell(\mathbf{X}) - \langle \boldsymbol{\lambda}_\ell - \boldsymbol{\nu}_\ell, \mathbf{t}(\mathbf{X})\rangle) \pi(\ell \mid \mathbf{X}) + \frac{1}{\beta}\Psi(\pi(\cdot \mid \mathbf{X}))\right] \\
- \sum_{\ell \in \llbracket L \rrbracket} \langle \boldsymbol{\lambda}_\ell + \boldsymbol{\nu}_\ell, \boldsymbol{\varepsilon}\rangle.
\tag{16}
$$

Thus, using the variational representation of $\mathrm{LSE}_\beta$, recalled in Lemma F.1, we have

$$
\min_\pi \mathcal{L}(\pi, \mathbf{\Lambda}, \mathbf{V}) = -\max_\pi \left\{ \mathbb{E}_{\mathbf{X}}\left[\sum_{\ell \in \llbracket L \rrbracket} (\langle \boldsymbol{\lambda}_\ell - \boldsymbol{\nu}_\ell, \mathbf{t}(\mathbf{X})\rangle - r_\ell(\mathbf{X})) \pi(\ell \mid \mathbf{X}) - \frac{1}{\beta}\Psi(\pi(\cdot \mid \mathbf{X}))\right] \right. \\
\left. + \sum_{\ell \in \llbracket L \rrbracket} \langle \boldsymbol{\lambda}_\ell + \boldsymbol{\nu}_\ell, \boldsymbol{\varepsilon}\rangle \right\} \\
= -\mathbb{E}_{\mathbf{X}}\left[\mathrm{LSE}_\beta\left((\langle \boldsymbol{\lambda}_\ell - \boldsymbol{\nu}_\ell, \mathbf{t}(\mathbf{X})\rangle - r_\ell(\mathbf{X}))_{\ell \in \llbracket L \rrbracket}\right)\right] - \sum_{\ell \in \llbracket L \rrbracket} \langle \boldsymbol{\lambda}_\ell + \boldsymbol{\nu}_\ell, \boldsymbol{\varepsilon}\rangle,
$$

and the optimum in the above problem for every $\mathbf{\Lambda}, \mathbf{V} \geqslant 0$ is achieved by $\pi_{\mathbf{\Lambda}, \mathbf{V}}$, defined in (7). Thus, we have

$$(\star) = \max_{\mathbf{\Lambda}, \mathbf{V} \geqslant 0} \{-F(\mathbf{\Lambda}, \mathbf{V})\} = \mathcal{R}_\beta(\pi_{\mathbf{\Lambda}^\star, \mathbf{V}^\star}). \tag{17}$$

The proof is concluded. ∎

***Proof of Lemma*** *3.2.* As shown in (15), for any randomized prediction function $\pi$, we can write

$$\mathcal{U}_s(\pi, \ell) = |\mathbb{E}_{\mathbf{X}}\left[\pi(\ell \mid \mathbf{X})t_s(\mathbf{X})\right]| .$$

Our goal is to show that $\pi^\star$ satisfies the required fairness constraints. We are going to rely on Lemma A.1 Subtracting the first equation in (14) from the second one, we deduce that $\gamma_\ell + \gamma'_\ell = 2\varepsilon$. Since $\gamma_\ell, \gamma'_\ell \geqslant 0$, then we conclude that $\gamma_{\ell s}, \gamma'_{\ell s} \in [0, 2\varepsilon_s]$. The above implies that

$$-\varepsilon_s \leqslant \mathbb{E}_{\mathbf{X}}\left[\pi^\star(\ell \mid \mathbf{X})t_s(\mathbf{X})\right] = \mathcal{U}_s(\pi^\star, \ell) \leqslant \varepsilon_s ,$$

as claimed. ∎

***Proof of Lemma*** *3.3.* Fix some randomized prediction function $\pi$ that is feasible for the problem in (3). In particular, it must be supported on $\widehat{\mathcal{Y}}$. Then, we can bound its negative risk as follows

$$
\begin{aligned}
-\mathcal{R}(\pi) &= -\mathbb{E}_{\mathbf{X}}\left[\sum_{\ell \in \llbracket L \rrbracket} r_\ell(\mathbf{X})\pi(\ell \mid \mathbf{X})\right] \\
&\overset{(a)}{\leqslant} \mathbb{E}_{\mathbf{X}}\left[\sum_{\ell \in \llbracket L \rrbracket} \left(\langle \boldsymbol{\lambda}_\ell^\star - \boldsymbol{\nu}_\ell^\star, \boldsymbol{t}(\mathbf{X})\rangle - r_\ell(\mathbf{X})\right)\pi(\ell \mid \mathbf{X})\right] + \sum_{\ell \in \llbracket L \rrbracket}\langle \boldsymbol{\lambda}_\ell^\star + \boldsymbol{\nu}_\ell^\star, \boldsymbol{\varepsilon}\rangle \\
&\overset{(b)}{\leqslant} \mathbb{E}_{\mathbf{X}}\left[\mathrm{LSE}_\beta\left((\langle \boldsymbol{\lambda}_\ell^\star - \boldsymbol{\nu}_\ell^\star, \boldsymbol{t}(\mathbf{X})\rangle - r_\ell(\mathbf{X}))_{\ell=-L}^{L}\right)\right] + \sum_{\ell \in \llbracket L \rrbracket}\langle \boldsymbol{\lambda}_\ell^\star + \boldsymbol{\nu}_\ell^\star, \boldsymbol{\varepsilon}\rangle \\
&\overset{(c)}{=} \mathbb{E}_{\mathbf{X}}\left[\sum_{\ell \in \llbracket L \rrbracket}\left(\langle \boldsymbol{\lambda}_\ell^\star - \boldsymbol{\nu}_\ell^\star, t(\mathbf{X})\rangle - r_\ell(\mathbf{X})\right)\pi^\star(\ell \mid \mathbf{X}) - \frac{1}{\beta}\Psi(\pi^\star(\cdot \mid \mathbf{X}))\right] + \sum_{\ell \in \llbracket L \rrbracket}\langle \boldsymbol{\lambda}_\ell^\star + \boldsymbol{\nu}_\ell^\star, \boldsymbol{\varepsilon}\rangle \\
&\overset{(d)}{\leqslant} \mathbb{E}_{\mathbf{X}}\left[\sum_{\ell \in \llbracket L \rrbracket}\left(\langle \boldsymbol{\lambda}_\ell^\star - \boldsymbol{\nu}_\ell^\star, \boldsymbol{t}(\mathbf{X})\rangle - r_\ell(\mathbf{X})\right)\pi^\star(\ell \mid \mathbf{X})\right] + \sum_{\ell \in \llbracket L \rrbracket}\langle \boldsymbol{\lambda}_\ell^\star + \boldsymbol{\nu}_\ell^\star, \boldsymbol{\varepsilon}\rangle + \frac{\log(2L+1)}{\beta} \\
&\overset{(e)}{=} -\mathcal{R}(\pi^\star) + \frac{\log(2L+1)}{\beta} ,
\end{aligned}
\tag{18}
$$

for (a) we used the assumption that $\pi$ is fair (*i.e.,* $\mathcal{U}_s(\pi, \ell) \leqslant \varepsilon_s$), which due to the fact that $\mathbf{\Lambda}^\star, \mathbf{V}^\star \geqslant 0$ implies

$$\sum_{\ell \in \llbracket L \rrbracket}\langle \boldsymbol{\lambda}_\ell^\star, \mathbb{E}_{\mathbf{X}}\left[\boldsymbol{t}(\mathbf{X})\pi(\ell \mid \mathbf{X})\right] + \boldsymbol{\varepsilon}\rangle + \sum_{\ell \in \llbracket L \rrbracket}\langle \boldsymbol{\nu}_\ell^\star, -\mathbb{E}_{\mathbf{X}}\left[\boldsymbol{t}(\mathbf{X})\pi(\ell \mid \mathbf{X})\right] + \boldsymbol{\varepsilon}\rangle \geqslant 0 ,$$

since every term in the summation is non-negative; (b) uses the fact that $\mathrm{LSE}_\beta(\boldsymbol{w}) \geqslant \langle \boldsymbol{w}, \boldsymbol{p}\rangle$ for any probability vector $\boldsymbol{p}$ (see Lemma F.1 for details); (c) relies on the variational representation of the $\mathrm{LSE}_\beta$, recalled in Lemma F.1 and the definition of $\pi^\star(\cdot \mid \mathbf{X})$; (d) uses the uniform bound on the entropy; (e) the last equality relies on the complementary slackness condition (14) of Lemma A.1. It ensures that

$$\begin{cases} \lambda_{\ell s}^\star \mathbb{E}_{\mathbf{X}}\left[\pi^\star(\ell \mid \mathbf{X})t_s(\mathbf{X})\right] = -\lambda_{\ell s}^\star \varepsilon_s \\ \nu_{\ell s}^\star \mathbb{E}_{\mathbf{X}}\left[\pi^\star(\ell \mid \mathbf{X})t_s(\mathbf{X})\right] = \nu_{\ell s}^\star \varepsilon_s \end{cases} \qquad \forall \ell \in \llbracket L \rrbracket, s \in [K] ,$$

implying that

$$\mathbb{E}_{\mathbf{X}}\left[\sum_{\ell \in \llbracket L \rrbracket}\langle \boldsymbol{\lambda}_\ell^\star - \boldsymbol{\nu}_\ell^\star, \boldsymbol{t}(\mathbf{X})\rangle \pi^\star(\ell \mid \mathbf{X})\right] + \sum_{\ell \in \llbracket L \rrbracket}\langle \boldsymbol{\lambda}_\ell^\star + \boldsymbol{\nu}_\ell^\star, \boldsymbol{\varepsilon}\rangle = 0 .$$

The proof is concluded. ∎

**Proof of Lemma 3.5.** Fix arbitrary $\mathbf{\Lambda}, \mathbf{V} \geqslant 0$ and consider $\pi_{\mathbf{\Lambda}, \mathbf{V}}$, defined in (7). To ease the notation, in this proof, we write $\pi$ instead of $\pi_{\mathbf{\Lambda}, \mathbf{V}}$.

**Part I.** Let us first recall the definition of the gradient map $\boldsymbol{G}_\alpha$ given in (8). We have the following expression

$$\boldsymbol{G}_\alpha\left(\mathbf{\Lambda}, \mathbf{V}\right) = \frac{(\mathbf{\Lambda}, \mathbf{V}) - ((\mathbf{\Lambda}, \mathbf{V}) - \alpha \nabla F\left(\mathbf{\Lambda}, \mathbf{V}\right))_+}{\alpha},$$

where $(\cdot)_+$ is to be understood entry-wise. Observing that for any $\alpha, a \geqslant 0$ and $b \in \mathbb{R}$, we have

$$\left| \frac{a - (a - \alpha b)_+}{\alpha} \right| = \left| \frac{a - \max\{0; a - \alpha b\}}{\alpha} \right| = \left| \min\left\{ \frac{a}{\alpha}; b \right\} \right| \geqslant |\min\{0; b\}| \geqslant (-b)_+,$$

we deduce that

$$\left\| (-\nabla F(\mathbf{\Lambda}, \mathbf{V}))_+ \right\| \leqslant \| \boldsymbol{G}_\alpha(\mathbf{\Lambda}, \mathbf{V}) \| \qquad \forall \mathbf{\Lambda}, \mathbf{V} \geqslant 0. \tag{19}$$

Relying on (10) and the expression for $\pi$ in (7), we observe that

$$\begin{aligned}
\nabla_{\lambda_{\ell s}} F(\mathbf{\Lambda}, \mathbf{V}) &= \mathbb{E}_{\boldsymbol{X}} \left[ \pi(\ell \mid \boldsymbol{X}) t_s(\boldsymbol{X}) \right] + \varepsilon_s \\
\nabla_{\nu_{\ell s}} F(\mathbf{\Lambda}, \mathbf{V}) &= -\mathbb{E}_{\boldsymbol{X}} \left[ \pi(\ell \mid \boldsymbol{X}) t_s(\boldsymbol{X}) \right] + \varepsilon_s
\end{aligned} \tag{20}$$

and that $\mathcal{U}_s(\pi, \ell) = |\mathbb{E}\left[ \pi(\ell \mid \boldsymbol{X}) t_s(\boldsymbol{X}) \right]|$ as it is shown in (15). Using the fact that $(|a| - c)_+^2 = (-a - c)_+^2 + (a - c)_+^2$ for all $a \in \mathbb{R}$ and $c \geqslant 0$, we deduce from above

$$(\mathcal{U}_s(\pi_{\mathbf{\Lambda}, \mathbf{V}}, \ell) - \varepsilon_s)_+^2 = (-\nabla_{\lambda_{\ell s}} F(\mathbf{\Lambda}, \mathbf{V}))_+^2 + (-\nabla_{\nu_{\ell s}} F(\mathbf{\Lambda}, \mathbf{V}))_+^2 \qquad \forall \ell \in [\![L]\!], s \in [K]. \tag{21}$$

Thus, we have shown

$$\sum_{\substack{\ell \in [\![L]\!] \\ s \in [K]}} (\mathcal{U}_s(\pi, \ell) - \varepsilon_s)_+^2 = \left\| (-\nabla F(\mathbf{\Lambda}, \mathbf{V}))_+ \right\|^2,$$

and (19) yields the claim.

**Part II.** We note that $\pi_{(\mathbf{\Lambda}, \mathbf{V})}$ is a unique solution to

$$\min_\pi \mathcal{L}(\pi, \mathbf{\Lambda}, \mathbf{V}),$$

where $\mathcal{L}$ is the Lagrangian defined in (16). Furthermore, $\min_\pi \mathcal{L}(\pi, \mathbf{\Lambda}, \mathbf{V}) = -F(\mathbf{\Lambda}, \mathbf{V}) = \mathcal{L}(\pi_{(\mathbf{\Lambda}, \mathbf{V})}, \mathbf{\Lambda}, \mathbf{V})$. Hence,

$$\begin{aligned}
\mathcal{R}_\beta(\pi_{(\mathbf{\Lambda}, \mathbf{V})}) + F(\mathbf{\Lambda}, \mathbf{V}) &= \sum_{\ell \in [\![L]\!], s \in [K]} \lambda_{\ell s} \nabla_{\lambda_{\ell s}} F(\mathbf{\Lambda}, \mathbf{V}) + \sum_{\ell \in [\![L]\!], s \in [K]} \nu_{\ell s} \nabla_{\nu_{\ell s}} F(\mathbf{\Lambda}, \mathbf{V}) \\
&= \langle (\mathbf{\Lambda}, \mathbf{V}), \nabla F(\mathbf{\Lambda}, \mathbf{V}) \rangle.
\end{aligned}$$

For the sake of simplicity let us denote by $\boldsymbol{u} \stackrel{\text{def}}{=} (\mathbf{\Lambda}, \mathbf{V}) \in \mathbb{R}^{2K(2L+1)}$ and recall the definition of gradient mapping $\boldsymbol{G}_\alpha$ given in (8). For any $j \in [2K(2L+1)]$ and $\alpha > 0$, we have

$$\boldsymbol{G}_{\alpha j}(\boldsymbol{u}) = \begin{cases} u_j / \alpha & \text{if } \alpha \partial_j F(\boldsymbol{u}) > u_j, \\ \partial_j F(\boldsymbol{u}) & \text{if } \partial_j F(\boldsymbol{u}) \leqslant u_j. \end{cases}$$

To bound $\langle \boldsymbol{u}, \nabla F(\boldsymbol{u}) \rangle$, let us examine each term of the scalar product. Denoting by $\widetilde{\boldsymbol{u}} \stackrel{\text{def}}{=} \boldsymbol{u} - \alpha \nabla F(\boldsymbol{u}) \in \mathbb{R}^{2K(2L+1)}$, for any $j \in [2K(2L+1)]$ and $\alpha > 0$, we have

$$u_j \partial_j F(\boldsymbol{u}) = \alpha \partial_j F(\boldsymbol{u}) \boldsymbol{G}_{\alpha j}(\boldsymbol{u}) \mathbb{I}\{\widetilde{u}_j < 0\} + u_j \boldsymbol{G}_{\alpha j}(\boldsymbol{u}) \mathbb{I}\{\widetilde{u}_j \geqslant 0\}.$$

Thus, it holds that

$$\begin{aligned}
\langle \boldsymbol{u}, \nabla F(\boldsymbol{u}) \rangle &\leqslant \sum_{j \in [2K(2L+1)]} (\alpha |\partial_j F(\boldsymbol{u}) \boldsymbol{G}_{\alpha j}(\boldsymbol{u})| + |u_j \boldsymbol{G}_{\alpha j}(\boldsymbol{u})|) \leqslant (\|\boldsymbol{u}\| + \alpha \|\nabla F(\boldsymbol{u})\|) \|\boldsymbol{G}_\alpha(\boldsymbol{u})\| \\
&\leqslant \left( \|\boldsymbol{u}\| + \alpha \sigma + \alpha \|\boldsymbol{\varepsilon}\| \sqrt{2(2L+1)} \right) \|\boldsymbol{G}_\alpha(\boldsymbol{u})\|,
\end{aligned}$$

where the last inequality follows from Lemma F.5 that bounds $\|\nabla F(\boldsymbol{u})\|$.

To conclude the proof, we use the fact that It remains to observe that $\min_{\mathbf{\Lambda}, \mathbf{V}} F(\mathbf{\Lambda}, \mathbf{V}) = -\mathcal{R}_\beta(\pi_{(\mathbf{\Lambda}^\star, \mathbf{V}^\star)})$, the bound (19) on $\|(-\nabla F(\mathbf{\Lambda}, \mathbf{V}))_+\|$ and the fact that $\mathcal{R}_\beta(\pi) \leqslant \mathcal{R}(\pi) + \frac{\log(2L+1)}{\beta}$ for all $\pi$. $\blacksquare$

# B    Bound on the variance of the stochastic gradient and its' smoothness

***Proof of Lemma*** *4.1*. We have

$$\mathbb{E}_{\boldsymbol{X}} \left\| g_{\boldsymbol{\Lambda},\mathbf{V}}(\boldsymbol{X}) - \nabla_{\boldsymbol{\Lambda},\mathbf{V}} F(\boldsymbol{\Lambda},\mathbf{V}) \right\|^2$$

$$\leqslant 2\mathbb{E}_{\boldsymbol{X}} \left\| \left( \sigma_\ell \left( \beta \left( \langle \lambda_{\ell'} - \nu_{\ell'}, t(\boldsymbol{X}) \rangle - r_{\ell'}(\boldsymbol{X}) \right)_{\ell'=-L}^{L} \right) t_s(\boldsymbol{X}) \right)_{\ell \in \llbracket L \rrbracket, s \in [K]} \right\|^2$$

$$\leqslant 2\mathbb{E}_{\boldsymbol{X}} \left[ \sum_{s \in [K]} t_s^2(\boldsymbol{X}) \right] \leqslant 2 \sum_{s \in [K]} \frac{1 - p_s}{p_s} \stackrel{\text{def}}{=} \sigma^2,$$

where the first inequality follows from the expressions of $g_{\boldsymbol{\Lambda},\mathbf{V}}$ and $\nabla_{\boldsymbol{\Lambda},\mathbf{V}} F(\boldsymbol{\Lambda},\mathbf{V})$, and the fact that $\mathrm{Var}(X + a) = \mathrm{Var}(X) \leqslant \mathbb{E}[X^2]$; the second inequality follows from the facts that $\|(a_i b_j)_{i,j}\|_2^2 = \|\boldsymbol{a}\|_2^2 \|\boldsymbol{b}\|_2^2 \leqslant \|\boldsymbol{a}\|_1^2 \|\boldsymbol{b}\|_2^2$ and that $\|\sigma(\cdot)\|_1 = 1$; and the last inequality follows from Lemma F.4.    ∎

***Proof of Lemma*** *3.4*. The goal of this proof is to show that the gradient of $(\boldsymbol{\Lambda},\mathbf{V}) \mapsto F(\boldsymbol{\Lambda},\mathbf{V})$ is $M$-Lipschitz. To this end, we first introduce some, rather heavy, but convenient, notation which will allow us to derive the announced result.

**Introducing notation.**    We first vectorize the variables $(\boldsymbol{\Lambda},\mathbf{V})$ and express them as

$$\boldsymbol{z} \stackrel{\text{def}}{=} (\underbrace{\lambda_{-L1}, \cdots \lambda_{-LK}}_{=\lambda_{-L}}, \cdots\cdots, \underbrace{\lambda_{L1}, \cdots \lambda_{LK}}_{=\lambda_L}, \underbrace{\nu_{-L1}, \cdots \nu_{-LK}}_{=\nu_{-L}}, \cdots\cdots, \underbrace{\nu_{L1}, \cdots \nu_{LK}}_{=\nu_L}) \in \mathbb{R}^{2K(2L+1)}.$$

Furthermore, for each $\boldsymbol{x} \in \mathbb{R}^d$, we introduce a matrix $\mathbf{A}(\boldsymbol{x}) \in \mathbb{R}^{(2L+1) \times 2K(2L+1)}$ defined as

$$\mathbf{A}(\boldsymbol{x}) \stackrel{\text{def}}{=} \begin{pmatrix} \boldsymbol{t}(\boldsymbol{x})^\top & 0\cdots 0 & \cdots & 0\cdots 0 & -\boldsymbol{t}(\boldsymbol{x})^\top & 0\cdots 0 & \cdots & 0\cdots 0 \\ 0\cdots 0 & \boldsymbol{t}(\boldsymbol{x})^\top & \cdots & 0\cdots 0 & 0\cdots 0 & -\boldsymbol{t}(\boldsymbol{x})^\top & \cdots & 0\cdots 0 \\ \vdots & \vdots & \ddots & \vdots & \vdots & \vdots & \ddots & \vdots \\ 0\cdots 0 & 0\cdots 0 & \cdots & \boldsymbol{t}(\boldsymbol{x})^\top & 0\cdots 0 & 0\cdots 0 & \cdots & -\boldsymbol{t}(\boldsymbol{x})^\top \end{pmatrix},$$

as well as

$$\boldsymbol{b}(\boldsymbol{x}) \stackrel{\text{def}}{=} (r_{-L}(\boldsymbol{X}), \cdots, r_L(\boldsymbol{X}))^\top \in \mathbb{R}^{2L+1} \qquad \text{and}$$

$$\boldsymbol{c} \stackrel{\text{def}}{=} (\varepsilon_1, \cdots, \varepsilon_K, \varepsilon_1, \cdots, \varepsilon_K, \cdots\cdots, \varepsilon_1, \cdots, \varepsilon_K)^\top \in \mathbb{R}^{2K(2L+1)}.$$

**Hessian of $F$ in the introduced notation.**    With the above introduce notation, we can express the function $F$ as

$$F(\boldsymbol{\Lambda},\mathbf{V}) = F(\boldsymbol{z}) = \mathbb{E}_{\boldsymbol{X}} \left[ \mathrm{LSE}_\beta(\mathbf{A}(\boldsymbol{X})\boldsymbol{z} - \boldsymbol{b}(\boldsymbol{X})) \right] + \langle \boldsymbol{c}, \boldsymbol{z} \rangle.$$

That is, $F$ is obtained from the $\mathrm{LSE}_\beta$ by a point-wise affine transformation of the coordinates plus a linear term. Chain rule yields the following expressions for the Hessian of $F$:

$$\nabla^2 F(\boldsymbol{z}) = \mathbb{E}_{\boldsymbol{X}} \left[ \mathbf{A}(\boldsymbol{X})^\top \nabla^2 \mathrm{LSE}_\beta(\mathbf{A}(\boldsymbol{X})\boldsymbol{z} - \boldsymbol{b}(\boldsymbol{X})) \mathbf{A}(\boldsymbol{X}) \right].$$

**Bounding the operator norm of the Hessian of $F$.**    To conclude the proof, we provide a uniform upper bound on the operator (spectral) norm of the Hessian of $F$. Using the Jensen's inequality and the fact that the operator norm is subordinate, we deduce that

$$\|\nabla^2 F(\boldsymbol{z})\|_{\mathrm{op}} = \|\mathbb{E}_{\boldsymbol{X}} \left[ \mathbf{A}(\boldsymbol{X})^\top \nabla^2 \mathrm{LSE}_\beta(\mathbf{A}(\boldsymbol{X})\boldsymbol{z} - \boldsymbol{b}(\boldsymbol{X})) \mathbf{A}(\boldsymbol{X}) \right]\|_{\mathrm{op}}$$

$$\leqslant \mathbb{E}_{\boldsymbol{X}} \left[ \|\mathbf{A}(\boldsymbol{X})^\top \nabla^2 \mathrm{LSE}_\beta(\mathbf{A}(\boldsymbol{X})\boldsymbol{z} - \boldsymbol{b}(\boldsymbol{X})) \mathbf{A}(\boldsymbol{X})\|_{\mathrm{op}} \right]$$

$$\leqslant \mathbb{E}_{\boldsymbol{X}} \left[ \|\mathbf{A}(\boldsymbol{X})\|_{\mathrm{op}} \|\nabla^2 \mathrm{LSE}_\beta(\mathbf{A}(\boldsymbol{X})\boldsymbol{z} - \boldsymbol{b}(\boldsymbol{X}))\|_{\mathrm{op}} \|\mathbf{A}(\boldsymbol{X})\|_{\mathrm{op}} \right].$$

Lemma F.2, implies that $\|\nabla^2 \mathrm{LSE}_\beta(\mathbf{A}(\boldsymbol{X})\boldsymbol{z} - \boldsymbol{b}(\boldsymbol{X}))\|_{\mathrm{op}} \leqslant \beta$ almost surely and for all $\boldsymbol{z}$. Thus, it remains to bound $\mathbb{E}_{\boldsymbol{X}} \|\mathbf{A}(\boldsymbol{X})\|_{\mathrm{op}}^2$ to conclude the proof. To this end, consider a vector $\boldsymbol{u}$, expressed "block-wise" as

$$\boldsymbol{u} = (\underbrace{u^\lambda_{-L1}, \cdots u^\lambda_{-LK}}_{\stackrel{\text{def}}{=} \boldsymbol{u}^\lambda_{-L}}, \cdots\cdots, \underbrace{u^\lambda_{L1}, \cdots u^\lambda_{LK}}_{\stackrel{\text{def}}{=} \boldsymbol{u}^\lambda_{L}}, \underbrace{u^\nu_{-L1}, \cdots u^\nu_{-LK}}_{\stackrel{\text{def}}{=} \boldsymbol{u}^\nu_{-L}}, \cdots\cdots, \underbrace{u^\nu_{L1}, \cdots u^\nu_{LK}}_{\stackrel{\text{def}}{=} \boldsymbol{u}^\nu_{L}})^\top \in \mathbb{R}^{2K(2L+1)}.$$

Using the definition of the operator norm and the expression for $\mathbf{A}(\boldsymbol{X})$, we deduce that

$$\mathbb{E}_{\boldsymbol{X}} \|\mathbf{A}(\boldsymbol{X})\|_{\mathrm{op}}^2 = \mathbb{E}_{\boldsymbol{X}} \sup_{\|\boldsymbol{u}\|_2^2=1} \|\mathbf{A}(\boldsymbol{X})\boldsymbol{u}\|_2^2$$

$$= \mathbb{E}_{\boldsymbol{X}} \sup_{\|\boldsymbol{u}\|_2^2=1} \sum_{\ell=-L}^{L} \left( \langle \boldsymbol{u}_\ell^\lambda - \boldsymbol{u}_\ell^\nu, \, t(\boldsymbol{X}) \rangle \right)^2$$

$$\leqslant 2\mathbb{E}_{\boldsymbol{X}} \left[ \|t(\boldsymbol{X})\|_2^2 \right] \sup_{\|\boldsymbol{u}\|_2^2=1} \sum_{\ell \in [\![L]\!]} \left( \|\boldsymbol{u}_\ell^\lambda\|_2^2 + \|\boldsymbol{u}_\ell^\nu\|_2^2 \right),$$

where the last inequality combines the Cauchy-Schwartz inequality and the fact that $\|\boldsymbol{v} - \boldsymbol{w}\|_2^2 \leqslant 2(\|\boldsymbol{v}\|_2^2 + \|\boldsymbol{w}\|_2^2)$ for all $\boldsymbol{v}, \boldsymbol{w} \in \mathbb{R}^m$. The proof is concluded using Lemma F.4 to bound $\mathbb{E}_{\boldsymbol{X}} \|t(\boldsymbol{X})\|_2^2$. ∎

**Lemma B.1** (Price of discretization). *Let Assumption 3.1 be satisfied. Let $\beta, B > 0, L \in \mathbb{N}$. Consider*

$$\mathcal{R}^\star \overset{\text{def}}{=} \inf_{h:\mathbb{R}^d \to \mathbb{R}} \left\{ \mathbb{E}(h(\boldsymbol{X}) - \eta(\boldsymbol{X}))^2 \ : \ \sup_{t \in \mathbb{R}} |\mathbb{P}(h(\boldsymbol{X}) \leqslant t \mid S = s) - \mathbb{P}(h(\boldsymbol{X}) \leqslant t)| \leqslant \varepsilon_s/2, \quad \forall s \in [K] \right\}.$$

*Then, it holds that*

$$\mathcal{R}(\pi_{\boldsymbol{\Lambda}^\star, \mathbf{V}^\star}) \leqslant \mathcal{R}^\star + \frac{4B}{L} + \frac{1}{L^2} + \frac{\log(2L+1)}{\beta}.$$

*Proof.* Let us assume that $\mathcal{R}^\star = \mathbb{E}(h^\star(\boldsymbol{X}) - \eta(\boldsymbol{X}))$ for some $h^\star : \mathbb{R}^d \to [-B, B]$. If it is not the case, the standard argument based on the minimizing sequence yields the same result.

Consider an operator $T_L$, which maps a deterministic classifier $h : \mathbb{R}^d \to [-B, B]$ onto a deterministic classifier $T_L(h) : \mathbb{R}^d \to \widehat{\mathcal{Y}}_L$, which is defined point-wise as follows

$$(T_L(h))(\boldsymbol{x}) = \lfloor Lh(\boldsymbol{x})/B \rfloor B/L \qquad \forall \boldsymbol{x} \in \mathbb{R}^d,$$

where $\lfloor a \rfloor$ is the closest integer smaller or equal to $a \in \mathbb{R}$ in absolute value. Notice, that for any $\ell \in \{-L, \dots, L-1\}$ and any $x \in \mathbb{R}^d$, we have

$$(T_L(h^\star))(\boldsymbol{x}) = \frac{\ell B}{L} \qquad \Longleftrightarrow \qquad h^\star(\boldsymbol{x}) \in \left[ \frac{\ell B}{L}, \frac{(\ell+1)B}{L} \right).$$

Moreover,

$$(T_L(h^\star))(\boldsymbol{x}) = B \qquad \Longleftrightarrow \qquad h^\star(\boldsymbol{x}) = B.$$

Since $h^\star$ satisfies $(\boldsymbol{\varepsilon}/2)$-fairness constraints, one checks that for all $\ell \in [\![L]\!], s \in [K]$

$$\mathcal{U}_s(T_L(h^\star), \ell) \leqslant \varepsilon_s.$$

That is, $T_L(h^\star)$ is feasible for the problem in (4). Therefore, Lemma 3.3 implies that

$$\mathcal{R}(\pi_{\boldsymbol{\Lambda}^\star, \mathbf{V}^\star}) \leqslant \mathcal{R}(T_L(h^\star)) + \frac{\log(2L+1)}{\beta}.$$

Furthermore, since $|T_L(h^\star)(\boldsymbol{x}) - h^\star(\boldsymbol{x})| \leqslant 1/L$ and $|\eta(\boldsymbol{x}) - h^\star(\boldsymbol{x})| \leqslant 2B$, we have

$$\mathcal{R}(T_L(h^\star)) = \mathbb{E} \left( \eta(\boldsymbol{X}) - T_L(h^\star)(\boldsymbol{X}) \right)^2 \leqslant \mathcal{R}(h^\star) + \frac{4B}{L} + \frac{1}{L^2}.$$

The proof is concluded. ∎

---

**Algorithm 2:** `AC-SA`$(F, \boldsymbol{w}_0, \mu, M, T)$

---

1: **Input:** function $F$; initial vector $\boldsymbol{w}_0$; parameters $\mu, M \geqslant 0$; number of iterations $T \geqslant 1$
2: $\boldsymbol{w}_0^{ag} = \boldsymbol{w}_0$
3: **for** $t = 1$ **to** $T$ **do**
4: $\quad$ sample new $z \sim P$, independently from the past
5: $\quad \alpha_t \leftarrow \frac{2}{t+1}$
6: $\quad \gamma_t \leftarrow \frac{4M}{t(t+1)}$
7: $\quad \boldsymbol{w}_t^{md} \leftarrow \frac{(1-\alpha_t)(\mu+\gamma_t)}{\gamma_t + (1-\alpha_t^2)\mu} \boldsymbol{w}_{t-1}^{ag} + \frac{\alpha_t((1-\alpha_t)\mu+\gamma_t)}{\gamma_t+(1-\alpha_t^2)\mu} \boldsymbol{w}_{t-1}$
8: $\quad \boldsymbol{w}_t \leftarrow \text{Proj}_W \left\{ \frac{(1-\alpha_t)\mu+\gamma_t}{\mu+\gamma_t} \boldsymbol{w}_{t-1} + \frac{\alpha_t\mu}{\mu+\gamma_t} \boldsymbol{w}_t^{md} - \frac{\alpha_t}{\mu+\gamma_t} \nabla f_{\boldsymbol{w}}(\boldsymbol{w}_t^{md}, z) \right\}$
9: $\quad \boldsymbol{w}_t^{ag} \leftarrow \alpha_t \boldsymbol{w}_t + (1-\alpha_t)\boldsymbol{w}_{t-1}^{ag}$
10: **end for**
11: **return** $\boldsymbol{w}_t^{ag}$

---

## C  Additional details on the algorithm

In this part of the appendix, we provide the analysis for the proposed algorithm. First, we introduce required notation and recall a result of Foster et al. (2019), who provided an algorithm for convex stochastic optimization. The provided algorithm is a refined version of the SDG3 algorithm of Allen-Zhu (2021). We note that Foster et al. (2019) give a control of the expected norm of a gradient, while we require a control of the expected squared norm of the gradient mapping. We introduce projection to the algorithm of Foster et al. (2019) based on the original algorithm of Ghadimi and Lan (2012) and provide a control of the expected squared norm of the gradient mapping of the final estimated solution.

**The setup and notation.**

Consider $f : \mathbb{R}^d \times \mathcal{Z} \to \mathbb{R}$, such that $\boldsymbol{w} \mapsto f(\boldsymbol{w}, z)$ is convex for each $z \in \mathcal{Z}$. Let $W \subset \mathbb{R}^d$ be a closed convex set. Let

$$F(\boldsymbol{w}) \stackrel{\text{def}}{=} \int f(\boldsymbol{w}, z) \mathrm{d} P(z)$$

for some probability distribution $P$ on $\mathcal{Z}$. In what follows, we assume that

$$\boldsymbol{w}^\star \in \arg\min_{\boldsymbol{w} \in W} F(\boldsymbol{w})$$

always exists.

**Assumption C.1.** *We assume that $F$ is $M$-smooth and the variance of $\nabla_{\boldsymbol{w}} f(\boldsymbol{w}, z)$ is bounded. That is, for some $M > 0$ and $\sigma > 0$*

$$\forall \boldsymbol{w}, \boldsymbol{w}' \in W \qquad \|\nabla F(\boldsymbol{w}) - \nabla F(\boldsymbol{w}')\| \leqslant M \|\boldsymbol{w} - \boldsymbol{w}'\| \qquad \text{and}$$

$$\forall \boldsymbol{w} \in W \qquad \int \left[ \|\nabla_{\boldsymbol{w}} f(\boldsymbol{w}, z) - \nabla F(\boldsymbol{w})\|^2 \right] \mathrm{d} P(z) \leqslant \sigma^2 .$$

Let us also define gradient mapping as

$$\boldsymbol{G}_{F,\alpha}(\boldsymbol{w}) \stackrel{\text{def}}{=} \frac{\boldsymbol{w} - \boldsymbol{w}_+}{\alpha} \quad \text{with} \quad \boldsymbol{w}_+ \in \arg\min_{\boldsymbol{w}' \in W} \left\{ \langle \nabla F(\boldsymbol{w}), \boldsymbol{w}' \rangle + \frac{1}{2\alpha} \|\boldsymbol{w}' - \boldsymbol{w}\|^2 \right\} .$$

Let $\text{Proj}_W(\cdot)$ be the Euclidean projection onto closed convex $W$.

### C.1  Some known results.

We start by introducing the original `AC-SA` algorithm of Ghadimi and Lan (2012) and recall some of their results for the sake of completeness.

**Algorithm 3:** $\texttt{AC-SA}^2(F, \boldsymbol{w}_0, \mu, M, T)$

---

1: **Input:** function $F$; initial vector $\boldsymbol{w}_0$; parameters $\mu, M \geqslant 0$; number of iterations $T \geqslant 1$
2: $\boldsymbol{w}_1 \leftarrow \texttt{AC-SA}(F, \boldsymbol{w}_0, \mu, M, \frac{T}{2})$
3: $\boldsymbol{w}_2 \leftarrow \texttt{AC-SA}(F, \boldsymbol{w}_1, \mu, M, \frac{T}{2})$
4: **return** $\boldsymbol{w}_2$

---

**Algorithm 4:** $\texttt{SGD3-refined}(F, \boldsymbol{w}_0, \mu, M, T)$

---

1: **Input:** function $F$; initial vector $\boldsymbol{w}_0$; parameters $0 < \mu \leqslant M$; number of iterations
   $T \geqslant \Omega\left(\frac{M}{\mu} \log_2 \frac{M}{\mu}\right)$
2: $F^{(0)}(\boldsymbol{w}) \leftarrow F(\boldsymbol{w}) + \frac{\mu}{2} \|\boldsymbol{w} - \boldsymbol{w}_0\|^2$ ; $\widehat{\boldsymbol{w}}_0 \leftarrow \boldsymbol{w}_0$; $\mu_0 \leftarrow \mu$
3: **for** $j = 1$ **to** $J = \left\lfloor \log \frac{M}{\mu} \right\rfloor$ **do**
4: $\quad \widehat{\boldsymbol{w}}_j \leftarrow \texttt{AC-SA}^2(F^{(j-1)}, \widehat{\boldsymbol{w}}_{j-1}, \mu_{j-1}, 2(M+\mu), \frac{T}{J})$
5: $\quad \mu_j \leftarrow 2\mu_{j-1}$
6: $\quad F^{(j)}(\boldsymbol{w}) \stackrel{\text{def}}{=} F^{(j-1)}(\boldsymbol{w}) + \frac{\mu_j}{2} \|\boldsymbol{w} - \widehat{\boldsymbol{w}}_j\|^2$
7: **end for**
8: **return** $\widehat{\boldsymbol{w}}_J$

---

**Theorem C.1.** *(Ghadimi and Lan, 2012, Proposition 9) Let $\boldsymbol{w}^\star \in \arg\min_{\boldsymbol{w} \in W} F(\boldsymbol{w})$, $\boldsymbol{w}_0 \in W$ a starting vector. If $F$ is $\mu-$strongly convex and $T \geqslant 1$ then with*

$$\alpha_t = \frac{2}{t+1} \quad \text{and} \quad \gamma_t = \frac{4M}{t(t+1)}, \quad \forall t > 1$$

*$\texttt{AC-SA}(F, \boldsymbol{w}_0, \mu, M, T)$, defined in Algorithm 2, outputs $\widehat{\boldsymbol{w}}_T$ satisfying*

$$\mathbb{E}[F(\widehat{\boldsymbol{w}}_T)] - F(\boldsymbol{w}^*) \leqslant \frac{2M \|\boldsymbol{w}_0 - \boldsymbol{w}^\star\|^2}{T^2} + \frac{8\sigma^2}{\mu T}.$$

Foster et al. (2019) propose another version of AC-SA, called $\texttt{AC-SA}^2$, which resets the stepsize halfway through the process.

**Lemma C.1.** *(Foster et al., 2019, Lemma 1) Let $W = \mathbb{R}^d$, $\boldsymbol{w}^\star \in \arg\min_{\boldsymbol{w} \in W} F(\boldsymbol{w})$, $\boldsymbol{w}_0 \in \mathbb{R}^d$ a starting vector. If $\mu > 0$, $M \geqslant 0$ and $T \geqslant 1$ then $\texttt{AC-SA}^2(F, \boldsymbol{w}_0, \mu, M, T)$, defined in Algorithm 3, outputs $\widehat{\boldsymbol{w}}$ satisfying*

$$\mathbb{E}[F(\widehat{\boldsymbol{w}})] - F(\boldsymbol{w}^*) \leqslant \frac{128M^2 \|\boldsymbol{w}_0 - \boldsymbol{w}^\star\|^2}{\mu T^4} + \frac{256M\sigma^2}{\mu^2 T^3} + \frac{16\sigma^2}{\mu T}.$$

**Remark C.1.** *Foster et al. (2019) do not consider constrained optimization throughout their work. However, the proof of Lemma C.1 follows analogous arguments.*

Foster et al. (2019) also introduce a refined version of algorithm SGD3 of Allen-Zhu (2021).

In what follows, we will show that Algorithm 4, after $T$ evaluations of the stochastic gradient, produces a point $\widehat{\boldsymbol{w}}$ such that $\mathbb{E}\|\boldsymbol{G}_{F,\alpha}(\widehat{\boldsymbol{w}})\|^2$ is controlled. This is a, rather mild, extension of Foster et al. (2019) and Allen-Zhu (2021).

## C.2 Control of the expected squared norm

Most of the proof techniques are already present in the original contribution of Allen-Zhu (2021) and Foster et al. (2019), we slightly extend their proof, introducing modifications related to the control of the squared norm and the projection step. For some $J \geqslant 1$, to be fixed later on, introduce

$$F_{\widetilde{\mu}}(\boldsymbol{w}) \stackrel{\text{def}}{=} F(\boldsymbol{w}) + \sum_{j=1}^{J} \frac{\mu_j}{2} \|\boldsymbol{w} - \widehat{\boldsymbol{w}}_j\|^2 \quad \text{and} \quad \boldsymbol{w}_{\widetilde{\mu}}^\star \in \arg\min_{\boldsymbol{w} \in W} F_{\widetilde{\mu}}(\boldsymbol{w}). \tag{22}$$

By construction, $F_{\widetilde{\mu}}$ is $\widetilde{\mu} \stackrel{\text{def}}{=} \sum_{j=1}^{J} \mu_j$-strongly convex and $(M + \widetilde{\mu})$-smooth. Let us also define $F^{(0)} \stackrel{\text{def}}{=} F(\boldsymbol{w})$ and $F^{(j)}(\boldsymbol{w}) \stackrel{\text{def}}{=} F^{(j-1)}(\boldsymbol{w}) + \frac{\mu_j}{2} \|\boldsymbol{w} - \widehat{\boldsymbol{w}}_j\|^2$, for $j = 1, 2, \ldots, J$. We will use the following results of Allen-Zhu (2021).

**Lemma C.2.** *(Allen-Zhu, 2021, Lemma 2.3) Let $\widetilde{F}$ be an $\widetilde{M}$-smooth and $\widetilde{\mu}$-strongly convex function. Let $\boldsymbol{w}, \boldsymbol{w}' \in W$ and $\boldsymbol{w}^+ = \boldsymbol{w} - \alpha \cdot \boldsymbol{G}_{\widetilde{F}, \alpha}(\boldsymbol{w})$. For any $\alpha \in \left(0, \frac{1}{\widetilde{M}}\right]$, we have*

$$\widetilde{F}(\boldsymbol{w}') \geqslant \widetilde{F}(\boldsymbol{w}^+) + \left\langle \boldsymbol{G}_{\widetilde{F}, \alpha}(\boldsymbol{w}), \, \boldsymbol{w}' - \boldsymbol{w} \right\rangle + \frac{\alpha}{2} \left\| \boldsymbol{G}_{\widetilde{F}, \alpha}(\boldsymbol{w}) \right\|^2 + \frac{\widetilde{\mu}}{2} \|\boldsymbol{w}' - \boldsymbol{w}\|^2 .$$

**Lemma C.3.** *(Allen-Zhu, 2021, Lemma 5.1) Consider $F_{\widetilde{\mu}}$ and $\boldsymbol{w}_{\widetilde{\mu}}^\star$ as defined in (22) and $\boldsymbol{w} \in W$. For any $\alpha \in \left(0, \frac{1}{M+\widetilde{\mu}}\right]$, we have*

$$\left\| \boldsymbol{G}_{F, \alpha}(\boldsymbol{w}) \right\| \leqslant \sum_{j=1}^{J} \mu_j \left\| \boldsymbol{w}_{\widetilde{\mu}}^\star - \widehat{\boldsymbol{w}}_j \right\| + 3 \left\| \boldsymbol{G}_{F_{\widetilde{\mu}}, \alpha}(\boldsymbol{w}) \right\| .$$

**Claim C.1.** *(Allen-Zhu, 2021, Claim 6.2) Suppose for every $j = 1, \ldots, J$ the iterates $\widehat{\boldsymbol{w}}_j$ of Algorithm 4 satisfy*

$$\mathbf{E}\left[ F^{(j-1)}\left(\widehat{\boldsymbol{w}}_j\right) - F^{(j-1)}\left(\boldsymbol{w}_{j-1}^\star\right) \right] \leqslant \delta_j \quad where \quad \boldsymbol{w}_{j-1}^\star \in \arg\min_{\boldsymbol{w}} \left\{ F^{(j-1)}(\boldsymbol{w}) \right\},$$

*then,*

(a) *for every $j \geqslant 1$ we have $\mathbf{E}\left[ \|\widehat{\boldsymbol{w}}_j - \boldsymbol{w}_{j-1}^\star\| \right]^2 \leqslant \mathbf{E}\left[ \|\widehat{\boldsymbol{w}}_j - \boldsymbol{w}_{j-1}^\star\|^2 \right] \leqslant \frac{2\delta_j}{\mu_{j-1}}$;*

(b) *for every $j \geqslant 1$ we have $\mathbf{E}\left[ \|\widehat{\boldsymbol{w}}_j - \boldsymbol{w}_j^\star\| \right]^2 \leqslant \mathbf{E}\left[ \|\widehat{\boldsymbol{w}}_j - \boldsymbol{w}_j^\star\|^2 \right] \leqslant \frac{\delta_j}{\mu_j}$;*

(c) *if $\mu_j = 2\mu_{j-1}$, then for all $j \geqslant 1$ we have $\mathbf{E}\left[ \sum_{j=1}^{J} \mu_j \|\widehat{\boldsymbol{w}}_j - \boldsymbol{w}_J^\star\| \right] \leqslant 4 \sum_{j=1}^{J} \sqrt{\delta_j \mu_j}$.*

In addition to Claim C.1, we prove the following lemma.

**Lemma C.4.** *Suppose for every $j = 1, \ldots, J$, $\mu_j = 2\mu_{j-1}$ and the iterates $\widehat{\boldsymbol{w}}_j$ of Algorithm 4 satisfy*

$$\mathbf{E}\left[ F^{(j-1)}\left(\widehat{\boldsymbol{w}}_j\right) - F^{(j-1)}\left(\boldsymbol{w}_{j-1}^\star\right) \right] \leqslant \delta_j \quad where \quad \boldsymbol{w}_{j-1}^\star \in \arg\min_{\boldsymbol{w}} \left\{ F^{(j-1)}(\boldsymbol{w}) \right\},$$

*then,*

$$\mathbf{E}\left[ \left( \sum_{j=1}^{J} \mu_j \|\boldsymbol{w}_J^\star - \widehat{\boldsymbol{w}}_j\| \right)^2 \right] \leqslant 16 J \sum_{j=1}^{J} \mu_j \delta_j .$$

*Proof.* Let $P_j \stackrel{\text{def}}{=} \sum_{t=1}^{j} \mu_t \left\| \boldsymbol{w}_j^\star - \widehat{\boldsymbol{w}}_t \right\|$, yielding that $P_J = \sum_{j=1}^{J}(P_j - P_{j-1})$, with the agreement that $P_0 = 0$. Cauchy-Schwartz inequality gives

$$P_J^2 = \left( \sum_{j=1}^{J}(P_j - P_{j-1}) \right)^2 \leqslant J \sum_{j=1}^{J}(P_j - P_{j-1})^2 . \tag{23}$$

Since $P_j$ is non-decreasing, to bound the above quantity, it suffices to bound each increment of the form $P_j - P_{j-1}$. One can write

$$P_j - P_{j-1} \stackrel{(a)}{\leqslant} \mu_j \left\| \boldsymbol{w}_j^\star - \widehat{\boldsymbol{w}}_j \right\| + \sum_{t=1}^{j-1} \mu_t (\left\| \boldsymbol{w}_j^\star - \widehat{\boldsymbol{w}}_t \right\| - \left\| \boldsymbol{w}_{j-1}^\star - \widehat{\boldsymbol{w}}_t \right\|)$$

$$\stackrel{(b)}{\leqslant} \mu_j \left\| \boldsymbol{w}_j^\star - \widehat{\boldsymbol{w}}_j \right\| + \left( \sum_{t=1}^{j-1} \mu_t \right) \left\| \boldsymbol{w}_j^\star - \boldsymbol{w}_{j-1}^\star \right\|$$

$$\stackrel{(c)}{\leqslant} \mu_j (2 \left\| \boldsymbol{w}_j^\star - \widehat{\boldsymbol{w}}_j \right\| + \left\| \boldsymbol{w}_{j-1}^\star - \widehat{\boldsymbol{w}}_j \right\|),$$

where (a) follows from the definition of $P_j$, (b) from reverse triangle inequality and (c) uses triangle inequality and the fact that $\sum_{t=1}^{j-1} \mu_t \leqslant \mu_j$ as $\mu_j = 2\mu_{j-1}$. Therefore, using the fact that $(a+b)^2 \leqslant 2a^2 + 2b^2$, we deduce from the above that

$$(P_j - P_{j-1})^2 \leqslant 2\mu_j^2 (4\left\|\boldsymbol{w}_j^\star - \widehat{\boldsymbol{w}}_j\right\|^2 + \left\|\boldsymbol{w}_{j-1}^\star - \widehat{\boldsymbol{w}}_j\right\|^2).$$

Taking the expectation and applying Claim C.1(a) and Claim C.1(b), the latter is bounded as

$$\mathbf{E}\left[(P_j - P_{j-1})^2\right] \leqslant 8\mu_j^2 \mathbf{E}[\|\boldsymbol{w}_j^\star - \widehat{\boldsymbol{w}}_j\|^2] + 2\mu_j^2 \mathbf{E}[\|\boldsymbol{w}_{j-1}^\star - \widehat{\boldsymbol{w}}_j\|^2] \leqslant 8\mu_j^2 \frac{\delta_j}{\mu_j} + 2\mu_j^2 \frac{2\delta_j}{\mu_{j-1}} = 16\mu_j\delta_j.$$
(24)

Plugging (24) into (23) yields the claimed bound. ∎

**Remark C.2.** *Notice, that in Algorithm 4 we apply* AC-SA² *to* $F^{(j-1)}$ *with starting point* $\widehat{\boldsymbol{w}}_{j-1}$ *and* $T/J$ *iterations. Since* $F^{(j-1)}$ *is* $M + \sum_{t=1}^{j-1} \mu_t \leqslant 2M-$*smooth and* $\mu_{j-1}-$*strongly convex, applying Lemma C.1 and Claim C.1(b), we get* $\mathbf{E}\left[F^{(j-1)}(\widehat{\boldsymbol{w}}_j) - F^{(j-1)}(\boldsymbol{w}_{j-1}^\star)\right] \leqslant \delta_j$ *and*

$$\delta_j \leqslant \frac{128(2M)^2 \mathbf{E}\left\|\widehat{\boldsymbol{w}}_{j-1} - \boldsymbol{w}_{j-1}^*\right\|^2}{\mu_{j-1}(T/J)^4} + \frac{256(2M)\sigma^2}{\mu_{j-1}^2(T/J)^3} + \frac{16\sigma^2}{\mu_{j-1}(T/J)}$$

$$\leqslant \frac{2^9 M^2 \delta_{j-1}}{\mu_{j-1}^2(T/J)^4} + \frac{2^9 M\sigma^2}{\mu_{j-1}^2(T/J)^3} + \frac{2^4\sigma^2}{\mu_{j-1}(T/J)}.$$

We are in position to prove the main ingredient of this section.

**Theorem C.2** (Control of the expected squared norm). *Let* $\boldsymbol{w}^\star \in \arg\min_{\boldsymbol{w} \in W} F(\boldsymbol{w})$, $\boldsymbol{w}_0 \in \mathbb{R}^d$ *a starting vector. When* $\mu \in (0, M]$ *and* $T > 2^{11/4}\sqrt{\frac{M}{\mu}}\left\lfloor \log_2 \frac{M}{\mu}\right\rfloor$, *then for* $\alpha = \frac{1}{2^{J+2}\mu}$, *with* $J = \left\lfloor \log_2 \frac{M}{\mu}\right\rfloor$, SGD3-refined$(F, \boldsymbol{w}_0, \mu, M, T)$ *outputs* $\widehat{\boldsymbol{w}}$ *satisfying*

$$\mathbf{E}\left[\|\boldsymbol{G}_{F,\alpha}(\widehat{\boldsymbol{w}})\|^2\right] \leqslant \left(\frac{3^4 \cdot 2^{16} M^2}{T^4}\log_2^5 \frac{M}{\mu} + 2\mu^2\right)\|\boldsymbol{w}_0 - \boldsymbol{w}_\mu^*\|^2$$

$$+ \frac{3^4 \cdot 2^{17} M\sigma^2}{\mu T^3}\log_2^4 \frac{M}{\mu} + \frac{3^4 \cdot 2^{11}\sigma^2}{T}\log_2^3 \frac{M}{\mu}.$$

*Proof.* **Part I.** At first, let us assume that $F$ is $\mu_0$-strongly convex. Since the definition of $F$ satisfies the definition given in (22) with $J - 1$, applying Lemma C.3 and using the fact that $(a+b)^2 \leqslant 2a^2 + 2b^2$, we get that for any $\alpha \in \left(0, (M + \sum_{j=1}^{J-1}\mu_j)^{-1}\right]$, it holds that

$$\mathbf{E}\left[\|\boldsymbol{G}_{F,\alpha}(\widehat{\boldsymbol{w}}_J)\|^2\right] \leqslant \mathbf{E}\left[\left(3\left\|\boldsymbol{G}_{F^{(J-1)},\alpha}(\widehat{\boldsymbol{w}}_J)\right\| + \sum_{j=1}^{J-1}\mu_j\left\|\boldsymbol{w}_{J-1}^\star - \widehat{\boldsymbol{w}}_j\right\|\right)^2\right]$$

$$\leqslant 2\mathbf{E}\left[9\left\|\boldsymbol{G}_{F^{(J-1)},\alpha}(\widehat{\boldsymbol{w}}_J)\right\|^2 + \left(\sum_{j=1}^{J-1}\mu_j\left\|\boldsymbol{w}_{J-1}^\star - \widehat{\boldsymbol{w}}_j\right\|\right)^2\right].$$
(25)

Note, that due to definition of $\mu_j$, we have $\sum_{j=1}^{J-1}\mu_j \leqslant M$, thus, the derived inequality holds for any $\alpha \in \left(0, (2M)^{-1}\right] \subset \left(0, (M + \sum_{j=1}^{J-1}\mu_j)^{-1}\right]$. Lemma C.4 provides a control of the second term of (25). To control the first term, let us apply Lemma C.2 with $F^{(J-1)}$ and $\boldsymbol{w} = \boldsymbol{w}' = \widehat{\boldsymbol{w}}_J$, getting $\frac{\alpha}{2}\left\|\boldsymbol{G}_{F^{(J-1)},\alpha}(\widehat{\boldsymbol{w}}_J)\right\|^2 \leqslant F^{(J-1)}(\widehat{\boldsymbol{w}}_J) - F^{(J-1)}(\widehat{\boldsymbol{w}}_J^+) \leqslant F^{(J-1)}(\widehat{\boldsymbol{w}}_J) - F^{(J-1)}(\boldsymbol{w}_{J-1}^*), \forall \alpha \in (0, \frac{1}{2M}]$. Meaning, that $\left\|\boldsymbol{G}_{F^{(J-1)},\alpha}(\widehat{\boldsymbol{w}}_J)\right\|^2 \leqslant \frac{2\delta_J}{\alpha}$. Let us recall, that $J = \left\lfloor \log_2 \frac{M}{\mu_0}\right\rfloor$ and $\mu_J = 2^J\mu_0 \leqslant M \leqslant 2\mu_J$. Hence, choosing $\alpha = \frac{1}{4\mu_J}$ and substituting the derived bound into (25), we deduce that

$$\mathbf{E}\left[\|G_{F,\alpha}(\widehat{\boldsymbol{w}}_J)\|^2\right] \leqslant \frac{36\delta_J}{\alpha} + 32(J-1)\sum_{j=1}^{J-1}\mu_j\delta_j \leqslant 144J\sum_{j=1}^{J}\mu_j\delta_j.$$

Now, let us substitute the bound on $\delta_j$ from Remark C.2 and replicate the steps of Foster et al. (2019) to control the above. We get

$$\sum_{j=1}^{J} \mu_j \delta_j \leqslant \frac{2^{10} M^2 \left\| \boldsymbol{w}_0 - \boldsymbol{w}^* \right\|^2}{(T/J)^4} + \frac{2^{10} M \sigma^2}{\mu_0 (T/J)^3} + \frac{2^5 \sigma^2}{(T/J)} + \sum_{j=2}^{J} \left( \frac{2^{10} M^2 \delta_{j-1}}{\mu_{j-1}(T/J)^4} + \frac{2^{10} M \sigma^2}{\mu_{j-1}(T/J)^3} + \frac{2^5 \sigma^2}{(T/J)} \right)$$

$$\leqslant \frac{2^{10} M^2 \left\| \boldsymbol{w}_0 - \boldsymbol{w}^* \right\|^2 J^4}{T^4} + \frac{2^{10} M \sigma^2 J^3}{\mu_0 T^3} \sum_{j=1}^{J} \frac{1}{2^{j-1}} + \frac{2^5 \sigma^2 J^2}{T} + \frac{2^{10} M^2 J^4}{T^4} \sum_{j=2}^{J} \frac{\delta_{j-1}}{\mu_{j-1}}$$

$$\leqslant \frac{2^{10} M^2 \left\| \boldsymbol{w}_0 - \boldsymbol{w}^* \right\|^2 J^4}{T^4} + \frac{2^{11} M \sigma^2 J^3}{\mu_0 T^3} + \frac{2^5 \sigma^2 J^2}{T} + \frac{2^{10} M^2 J^4}{\mu_0^2 T^4} \sum_{j=1}^{J} \mu_j \delta_j \,,$$

where the last inequality comes from the facts that $\sum_{j=1}^{J} \frac{1}{2^{j-1}} \leqslant 2$ and $\sum_{j=2}^{J} \frac{\delta_{j-1}}{\mu_{j-1}} \leqslant \sum_{j=1}^{J} \frac{\delta_j}{\mu_j} \leqslant \frac{1}{\mu_0^2} \sum_{j=1}^{J} \mu_j \delta_j$. Rearranging the terms and multiplying both sides by $144 J$, we get

$$144 J \sum_{j=1}^{J} \mu_j \delta_j \leqslant \frac{9}{1 - \frac{2^{10} M^2 J^4}{\mu_0^2 T^4}} \left( \frac{2^{14} M^2 \left\| \boldsymbol{w}_0 - \boldsymbol{w}^* \right\|^2 J^5}{T^4} + \frac{2^{15} M \sigma^2 J^4}{\mu_0 T^3} + \frac{2^9 \sigma^2 J^3}{T} \right) \,.$$

Choosing $T > 2^{11/4} J \sqrt{\frac{M}{\mu_0}}$ ensures that $\frac{1}{1 - \frac{2^{10} M^2 J^4}{\mu_0^2 T^4}} \leqslant 2$. Finally, substituting the derived bounds and the value of $J = \left\lfloor \log_2 \frac{M}{\mu_0} \right\rfloor$, we conclude that

$$\mathbf{E} \left[ \left\| \boldsymbol{G}_{F,\alpha}(\widehat{\boldsymbol{w}}_J) \right\|^2 \right] \leqslant \frac{9 \cdot 2^{15} M^2 \left\| \boldsymbol{w}_0 - \boldsymbol{w}^* \right\|^2}{T^4} \log_2^5 \frac{M}{\mu_0} + \frac{9 \cdot 2^{16} M \sigma^2}{\mu_0 T^3} \log_2^4 \frac{M}{\mu_0} + \frac{9 \cdot 2^{10} \sigma^2}{T} \log_2^3 \frac{M}{\mu_0} \,. \tag{26}$$

**Part II.** When $F$ is not strongly convex, let $F_\mu(\boldsymbol{w}) \stackrel{\text{def}}{=} F(\boldsymbol{w}) + \frac{\mu}{2} \left\| \boldsymbol{w} - \boldsymbol{w}_0 \right\|^2$ and $\boldsymbol{w}_\mu^\star \in \arg\min_{\boldsymbol{w}} \{ F_\mu(\boldsymbol{w}) \}$. Applying (26) and Lemma C.3 with $J = 1$ and $\widehat{\boldsymbol{w}}_1 = \boldsymbol{w}_0$, we get

$$\mathbf{E} \left[ \left\| \boldsymbol{G}_{F,\alpha}(\widehat{\boldsymbol{w}}) \right\|^2 \right] \leqslant \left( \frac{3^4 \cdot 2^{16} M^2}{T^4} \log_2^5 \frac{M}{\mu} + 2\mu^2 \right) \left\| \boldsymbol{w}_0 - \boldsymbol{w}_\mu^* \right\|^2$$

$$+ \frac{3^4 \cdot 2^{17} M \sigma^2}{\mu T^3} \log_2^4 \frac{M}{\mu} + \frac{3^4 \cdot 2^{11} \sigma^2}{T} \log_2^3 \frac{M}{\mu} \,.$$

Since $\frac{\mu}{2} \left\| \boldsymbol{w}^\star - \boldsymbol{w}_0 \right\|^2 - \frac{\mu}{2} \left\| \boldsymbol{w}_\mu^\star - \boldsymbol{w}_0 \right\|^2 = (F_\mu(\boldsymbol{w}^\star) - F(\boldsymbol{w}^\star)) + (F(\boldsymbol{w}_\mu^\star) - F_\mu(\boldsymbol{w}_\mu^\star)) \geqslant 0$, then $\left\| \boldsymbol{w}_\mu^\star - \boldsymbol{w}_0 \right\|^2 \leqslant \left\| \boldsymbol{w}^\star - \boldsymbol{w}_0 \right\|^2$. The proof is concluded. $\blacksquare$

**Remark C.3.** *Notice, that in Algorithm 4 we apply* `AC-SA` *to* $F^{(j-1)}$ *with starting point* $\widehat{\boldsymbol{w}}_{j-1}$ *and* $T/J$ *iterations. Since* $F^{(j-1)}$ *is* $M + \sum_{t=1}^{j-1} \mu_t \leqslant 2M-$*smooth and* $\mu_{j-1}-$*strongly convex, applying Lemma C.1 and Claim C.1(b), we get* $\mathbf{E} \left[ F^{(j-1)} \left( \widehat{\boldsymbol{w}}_j \right) - F^{(j-1)} \left( \boldsymbol{w}_{j-1}^\star \right) \right] \leqslant \delta_j$ *and*

$$\delta_j \leqslant \frac{2(2M) \mathbf{E} \left\| \widehat{\boldsymbol{w}}_{j-1} - \boldsymbol{w}_{j-1}^* \right\|^2}{(T/J)^2} + \frac{8\sigma^2}{\mu_{j-1}(T/J)} \leqslant \frac{4M \delta_{j-1}}{\mu_{j-1}(T/J)^2} + \frac{8\sigma^2}{\mu_{j-1}(T/J)} \,.$$

**Theorem C.3** (Control of the expected squared norm). *Let* $\boldsymbol{w}^\star \in \arg\min_{\boldsymbol{w} \in W} F(\boldsymbol{w})$, $\boldsymbol{w}_0 \in \mathbb{R}^d$ *a starting vector. When* $\mu \in (0, M]$ *and* $T > 4\sqrt{\frac{M}{\mu}} \left\lfloor \log_2 \frac{M}{\mu} \right\rfloor$, *then for* $\alpha = \frac{1}{2^{J+2}\mu}$, *with* $J = \left\lfloor \log_2 \frac{M}{\mu} \right\rfloor$, `SGD3-refined`$(F, \boldsymbol{w}_0, \mu, M, T)$ *with* `AC-SA` *outputs* $\widehat{\boldsymbol{w}}$ *satisfying*

$$\mathbf{E} \left[ \left\| \boldsymbol{G}_{F,\alpha}(\widehat{\boldsymbol{w}}) \right\|^2 \right] \leqslant \left( \frac{3^4 2^9 M \mu}{T^2} \log_2^3 \frac{M}{\mu} + 2\mu^2 \right) \left\| \boldsymbol{w}_0 - \boldsymbol{w}^* \right\|^2 + \frac{3^4 2^{11} \sigma^2}{T} \log_2^3 \frac{M}{\mu} \,.$$

*Proof.* **Part I.** At first, let us assume that $F$ is $\mu_0$-strongly convex. Applying Lemma C.3 and using the fact that $(a+b)^2 \leqslant 2a^2 + 2b^2$, we get

$$
\mathbf{E}\left[\|\boldsymbol{G}_{F,\alpha}(\widehat{\boldsymbol{w}}_J)\|^2\right] \leqslant \mathbf{E}\left[\left(3\left\|\boldsymbol{G}_{F^{(J-1)},\alpha}(\widehat{\boldsymbol{w}}_J)\right\| + \sum_{j=1}^{J-1}\mu_j\left\|\boldsymbol{w}_{J-1}^\star - \widehat{\boldsymbol{w}}_j\right\|\right)^2\right]
$$

$$
\leqslant 2\mathbf{E}\left[9\left\|\boldsymbol{G}_{F^{(J-1)},\alpha}(\widehat{\boldsymbol{w}}_J)\right\|^2 + \left(\sum_{j=1}^{J-1}\mu_j\left\|\boldsymbol{w}_{J-1}^\star - \widehat{\boldsymbol{w}}_j\right\|\right)^2\right]. \tag{27}
$$

Lemma C.4 provides a control of the second term of the above inequality. To control the first term, let us apply Lemma C.2 with $F^{(J-1)}$ and $\boldsymbol{w} = \boldsymbol{w}' = \widehat{\boldsymbol{w}}_J$, getting $\frac{\alpha}{2}\left\|\boldsymbol{G}_{F^{(J-1)},\alpha}(\widehat{\boldsymbol{w}}_J)\right\|^2 \leqslant F^{(J-1)}(\widehat{\boldsymbol{w}}_J) - F^{(J-1)}(\widehat{\boldsymbol{w}}_J^+) \leqslant F^{(J-1)}(\widehat{\boldsymbol{w}}_J) - F^{(J-1)}(\boldsymbol{w}_{J-1}^*), \forall \alpha \in (0, \frac{1}{2M}]$. Meaning, that $\left\|\boldsymbol{G}_{F^{(J-1)},\alpha}(\widehat{\boldsymbol{w}}_J)\right\|^2 \leqslant \frac{2\delta_J}{\alpha}$. Let us recall, that $J = \left\lfloor\log_2\frac{M}{\mu_0}\right\rfloor$ and $\mu_J = 2^J\mu_0 \leqslant M \leqslant 2\mu_J$. Hence, choosing $\alpha = \frac{1}{4\mu_J}$ and substituting the derived bound into (27), we deduce that

$$
\mathbf{E}\left[\|G_{F,\alpha}(\widehat{\boldsymbol{w}}_J)\|^2\right] \leqslant \frac{36\delta_J}{\alpha} + 32(J-1)\sum_{j=1}^{J-1}\mu_j\delta_j \leqslant 144J\sum_{j=1}^{J}\mu_j\delta_j.
$$

Let us substitute the bound on $\delta_j$ from Remark C.2 to control the above. We get

$$
\sum_{j=1}^{J}\mu_j\delta_j \leqslant \frac{4M\|\boldsymbol{w}_0 - \boldsymbol{w}^*\|^2\mu_1}{(T/J)^2} + \frac{8\sigma^2\mu_1}{(T/J)\mu_0} + \sum_{j=2}^{J}\left(\frac{8M\delta_{j-1}}{(T/J)^2} + \frac{16\sigma^2}{(T/J)}\right)
$$

$$
\leqslant \frac{8M\mu_0\|\boldsymbol{w}_0 - \boldsymbol{w}^*\|^2 J^2}{T^2} + \frac{32\sigma^2 J^2}{T} + \frac{8MJ^2}{T^2}\sum_{j=2}^{J}\delta_{j-1}
$$

$$
\leqslant \frac{8M\mu_0\|\boldsymbol{w}_0 - \boldsymbol{w}^*\|^2 J^2}{T^2} + \frac{32\sigma^2 J^2}{T} + \frac{8MJ^2}{\mu_0 T^2}\sum_{j=2}^{J}\mu_{j-1}\delta_{j-1}
$$

$$
\leqslant \frac{8M\mu_0\|\boldsymbol{w}_0 - \boldsymbol{w}^*\|^2 J^2}{T^2} + \frac{32\sigma^2 J^2}{T} + \frac{8MJ^2}{\mu_0 T^2}\sum_{j=1}^{J}\mu_j\delta_j.
$$

Rearranging the terms and multiplying both sides by $144J$, we get

$$
144J\sum_{j=1}^{J}\mu_j\delta_j \leqslant \frac{1152}{1 - \frac{8MJ^2}{\mu_0 T^2}}\left(\frac{M\mu_0\|\boldsymbol{w}_0 - \boldsymbol{w}^*\|^2 J^3}{T^2} + \frac{4\sigma^2 J^3}{T}\right).
$$

Choosing $T > 4J\sqrt{\frac{M}{\mu_0}}$ ensures that $\frac{1}{1 - \frac{8MJ^2}{\mu_0 T^2}} \leqslant 2$. Finally, substituting the derived bounds and the value of $J = \left\lfloor\log_2\frac{M}{\mu_0}\right\rfloor$, we conclude that

$$
\mathbf{E}\left[\|\boldsymbol{G}_{F,\alpha}(\widehat{\boldsymbol{w}}_J)\|^2\right] \leqslant 2304\log_2^3\frac{M}{\mu_0}\left(\frac{M\mu_0\|\boldsymbol{w}_0 - \boldsymbol{w}^*\|^2}{T^2} + \frac{4\sigma^2}{T}\right). \tag{28}
$$

**Part II.** When $F$ is not strongly convex, let $F_\mu(\boldsymbol{w}) \stackrel{\text{def}}{=} F(\boldsymbol{w}) + \frac{\mu}{2}\|\boldsymbol{w} - \boldsymbol{w}_0\|^2$ and $\boldsymbol{w}_\mu^\star \in \arg\min_{\boldsymbol{w}}\{F_\mu(\boldsymbol{w})\}$. Applying (26) and Lemma C.3 with $J = 1$ and $\widehat{\boldsymbol{w}}_1 = \boldsymbol{w}_0$, we get

$$
\mathbf{E}\left[\|\boldsymbol{G}_{F,\alpha}(\widehat{\boldsymbol{w}})\|^2\right] \leqslant \left(\frac{3^42^9 M\mu}{T^2}\log_2^3\frac{M}{\mu} + 2\mu^2\right)\|\boldsymbol{w}_0 - \boldsymbol{w}_\mu^*\|^2 + \frac{3^42^{11}\sigma^2}{T}\log_2^3\frac{M}{\mu}.
$$

Since $\frac{\mu}{2}\|\boldsymbol{w}^\star - \boldsymbol{w}_0\|^2 - \frac{\mu}{2}\|\boldsymbol{w}_\mu^\star - \boldsymbol{w}_0\|^2 = (F_\mu(\boldsymbol{w}^\star) - F(\boldsymbol{w}^\star)) + (F(\boldsymbol{w}_\mu^\star) - F_\mu(\boldsymbol{w}_\mu^\star)) \geqslant 0$, then $\|\boldsymbol{w}_\mu^\star - \boldsymbol{w}_0\|^2 \leqslant \|\boldsymbol{w}^\star - \boldsymbol{w}_0\|^2$. The proof is concluded. ∎

# D   Proofs of statistical guarantees

In order to derive statistical guarantees for the proposed method, we are going to instantiate the extension provided in the previous appendix.

***Proof of Theorem 5.1.*** Let us instantiate Theorem C.2. According to Lemma 4.1 and Lemma 3.4 we have that $\sigma^2 = 2 \sum_{s \in [S]} \frac{1 - p_s}{p_s}$ and $M = \beta \sigma^2$. Setting $\beta = \frac{T}{8 \log_2 T}$ and $\mu = \sigma^2 / \beta$, ensures that $\mu \leqslant M$ and that $T > 4 \sqrt{\frac{M}{\mu}} \left\lfloor \log_2 \frac{M}{\mu} \right\rfloor = \frac{T}{\log_2 T} \left\lfloor \log_2 \frac{T}{8 \log_2 T} \right\rfloor, \forall T \geqslant 2$. For $T$ larger than some large enough absolute constant, the conditions of Theorem C.2 are satisfied for the function $F$.

**Fairness guarantee.** Theorem C.2 yields

$$\mathbf{E}\left[\|\boldsymbol{G}_{F,\alpha}(\widehat{\boldsymbol{w}})\|^2\right] \leqslant \left(\frac{3^4 2^6 \sigma^4 \log_2^5 \frac{T^2}{64 \log_2^2 T}}{T^2} \frac{1}{\log_2^2 T} + \frac{2^7 \sigma^4}{T^2} \log_2^2 T\right) \|(\boldsymbol{\Lambda}^\star, \mathbf{V}^\star)\|^2$$

$$+ \frac{3^4 2^{11} \sigma^2}{T \log_2^2 T} \log_2^4 \frac{T^2}{64 \log_2^2 T} + \frac{3^4 2^{11} \sigma^2}{T} \log_2^3 \frac{T^2}{64 \log_2^2 T} \,.$$

Therefore, we have shown that

$$\mathbf{E}\left[\left\|\boldsymbol{G}_\alpha(\widehat{\boldsymbol{\Lambda}}, \widehat{\mathbf{V}})\right\|^2\right] \leqslant \widetilde{\mathcal{O}}\left(\frac{\sigma^2}{T}\left(1 + \frac{\sigma^2}{T}\|(\boldsymbol{\Lambda}^\star, \mathbf{V}^\star)\|^2\right)\right) \tag{29}$$

Hence, the first part of Lemma 3.5 implies the fairness guarantee.

**Fairness guarantee with `AC-SA`.** Theorem C.3 yields

$$\mathbf{E}\left[\|\boldsymbol{G}_{F,\alpha}(\widehat{\boldsymbol{w}})\|^2\right] \leqslant \left(\frac{3^4 2^9 \sigma^4}{T^2} \log_2^3 \frac{T^2}{64 \log_2^2 T} + \frac{2^7 \sigma^4}{T^2} \log_2^2 T\right) \|(\boldsymbol{\Lambda}^\star, \mathbf{V}^\star)\|^2$$

$$+ \frac{3^4 2^{11} \sigma^2}{T} \log_2^3 \frac{T^2}{64 \log_2^2 T} \,.$$

Therefore, we have shown that

$$\mathbf{E}\left[\left\|\boldsymbol{G}_\alpha(\widehat{\boldsymbol{\Lambda}}, \widehat{\mathbf{V}})\right\|^2\right] \leqslant \widetilde{\mathcal{O}}\left(\frac{\sigma^2}{T}\left(1 + \frac{\sigma^2}{T}\|(\boldsymbol{\Lambda}^\star, \mathbf{V}^\star)\|^2\right)\right) \,. \tag{30}$$

Hence, the first part of Lemma 3.5 implies the fairness guarantee.

**Risk guarantee.** The second part of Lemma 3.5 states that

$$\mathcal{R}(\pi_{\widehat{\boldsymbol{\Lambda}}, \widehat{\mathbf{V}}}) - \mathcal{R}(\pi_{\boldsymbol{\Lambda}^\star, \mathbf{V}^\star}) \leqslant \left(\left\|(\widehat{\boldsymbol{\Lambda}}, \widehat{\mathbf{V}})\right\| + \alpha\sigma + \alpha\|\boldsymbol{\varepsilon}\|\sqrt{2(2L+1)}\right) \cdot \left\|\boldsymbol{G}_\alpha(\widehat{\boldsymbol{\Lambda}}, \widehat{\mathbf{V}})\right\| + \frac{\log(2L+1)}{\beta} \,.$$

Taking the expectation and applying the Cauchy-Schwartz inequality, we obtain

$$\mathbf{E}\left[\mathcal{R}(\pi_{\widehat{\boldsymbol{\Lambda}}, \widehat{\mathbf{V}}})\right] - \mathcal{R}(\pi_{\boldsymbol{\Lambda}^\star, \mathbf{V}^\star}) \leqslant \left(\sqrt{\mathbf{E}\left[\left\|(\widehat{\boldsymbol{\Lambda}}, \widehat{\mathbf{V}})\right\|^2\right]} + \alpha\sigma + \alpha\|\boldsymbol{\varepsilon}\|\sqrt{2(2L+1)}\right)$$

$$\cdot \sqrt{\mathbf{E}\left[\left\|\boldsymbol{G}_\alpha(\widehat{\boldsymbol{\Lambda}}, \widehat{\mathbf{V}})\right\|^2\right]} + \frac{\log(2L+1)}{\sqrt{T}} \,.$$

Recalling the value of $\alpha = \frac{1}{2^{J+2}\mu_J} \leqslant \frac{1}{2M}$ from Theorems C.2 and C.3 and the fact that $M = \beta\sigma^2$, we get $\alpha\sigma \leqslant \frac{1}{2\beta\sigma}$. Finally, applying (29)-(30) and setting $L = \sqrt{T}$, we obtain

$$\mathbf{E}\left[\mathcal{R}(\pi_{\widehat{\boldsymbol{\Lambda}}, \widehat{\mathbf{V}}})\right] - \mathcal{R}(\pi_{\boldsymbol{\Lambda}^\star, \mathbf{V}^\star})$$

$$\leqslant \widetilde{\mathcal{O}}\left(\frac{\sigma}{\sqrt{T}}\left(1 + \frac{\sigma}{\sqrt{T}}\|(\boldsymbol{\Lambda}^\star, \mathbf{V}^\star)\|\right)\left(\mathbf{E}^{1/2}\left[\|(\widehat{\boldsymbol{\Lambda}}, \widehat{\mathbf{V}})\|^2\right] + \frac{1}{T\sigma} + \frac{\|\boldsymbol{\varepsilon}\|}{T^{3/4}\sigma}\right) + \frac{\log(\sqrt{T})}{\sqrt{T}}\right) \,.$$

Above combined with Lemma B.1 yields the claimed bound. ∎

# E   Unknown $\eta$ and $\tau$

In this section we consider the case, when $\eta$ and $\tau$ are unknown and estimated by $\widehat{\eta}$ and $\widehat{\tau}$. We denote by $\widehat{t}(x) \stackrel{\text{def}}{=} 1 - \frac{\widehat{\tau}(x)}{p}$ and $\widehat{r}_\ell(x) \stackrel{\text{def}}{=} \left(\widehat{\eta}(x) - \frac{\ell B}{L}\right)^2$. We consider the plug-in version of (5), defined as

$$\min_{\boldsymbol{\Lambda},\mathbf{V} \geqslant 0} \left\{ \mathbb{E}_{\boldsymbol{X}} \left[ \mathrm{LSE}_\beta \left( \left( \left\langle \boldsymbol{\lambda}_\ell - \boldsymbol{\nu}_\ell, \widehat{\boldsymbol{t}}(\boldsymbol{X}) \right\rangle - \widehat{r}_\ell(\boldsymbol{X}) \right)_{\ell=-L}^L \right) \right] + \sum_{\ell=-L}^L \langle \boldsymbol{\lambda}_\ell + \boldsymbol{\nu}_\ell, \boldsymbol{\varepsilon} \rangle \right\}. \quad (\widehat{\mathcal{P}}_{LSE})$$

Let us denote by $\widehat{F}$, the objective function of the above problem and introduce

$$\widehat{\mathcal{R}}_\beta(\pi) \stackrel{\text{def}}{=} \mathbb{E}_{\boldsymbol{X}} \left[ \sum_{\ell \in \llbracket L \rrbracket} \widehat{r}_\ell(\boldsymbol{X}) \pi(\ell \mid \boldsymbol{X}) + \frac{1}{\beta} \Psi(\pi(\cdot \mid \boldsymbol{X})) \right].$$

The gradient of $\widehat{F}$ is given for any $\boldsymbol{\Lambda}, \mathbf{V} \geqslant 0$ by

$$\nabla_{\lambda_{\ell_s}} \widehat{F}(\boldsymbol{\Lambda}, \mathbf{V}) = \mathbb{E}_{\boldsymbol{X}} \left[ \sigma_\ell \left( \beta \left( \left\langle \boldsymbol{\lambda}_{\ell'} - \boldsymbol{\nu}_{\ell'}, \widehat{\boldsymbol{t}}(\boldsymbol{X}) \right\rangle - \widehat{r}_{\ell'}(\boldsymbol{X}) \right)_{\ell'=-L}^L \right) \widehat{t}_s(\boldsymbol{X}) \right] + \varepsilon_s,$$

$$\nabla_{\nu_{\ell_s}} \widehat{F}(\boldsymbol{\Lambda}, \mathbf{V}) = -\mathbb{E}_{\boldsymbol{X}} \left[ \sigma_\ell \left( \beta \left( \left\langle \boldsymbol{\lambda}_{\ell'} - \boldsymbol{\nu}_{\ell'}, \widehat{\boldsymbol{t}}(\boldsymbol{X}) \right\rangle - \widehat{r}_{\ell'}(\boldsymbol{X}) \right)_{\ell'=-L}^L \right) \widehat{t}_s(\boldsymbol{X}) \right] + \varepsilon_s, \quad (31)$$

for $\ell \in \llbracket L \rrbracket, s \in [K]$. Let us denote by $\widehat{\boldsymbol{g}}(\boldsymbol{\Lambda}, \mathbf{V})$ the stochastic gradient of $\widehat{F}$, defined as

$$\widehat{g}_{\lambda_{\ell_s}}(\boldsymbol{\Lambda}, \mathbf{V}) = \sigma_\ell \left( \beta \left( \left\langle \boldsymbol{\lambda}_{\ell'} - \boldsymbol{\nu}_{\ell'}, \widehat{\boldsymbol{t}}(\boldsymbol{X}) \right\rangle - \widehat{r}_{\ell'}(\boldsymbol{X}) \right)_{\ell'=-L}^L \right) \widehat{t}_s(\boldsymbol{X}) + \varepsilon_s,$$

$$\widehat{g}_{\nu_{\ell_s}}(\boldsymbol{\Lambda}, \mathbf{V}) = -\sigma_\ell \left( \beta \left( \left\langle \boldsymbol{\lambda}_{\ell'} - \boldsymbol{\nu}_{\ell'}, \widehat{\boldsymbol{t}}(\boldsymbol{X}) \right\rangle - \widehat{r}_{\ell'}(\boldsymbol{X}) \right)_{\ell'=-L}^L \right) \widehat{t}_s(\boldsymbol{X}) + \varepsilon_s, \quad (32)$$

for $\boldsymbol{X} \sim \mathbb{P}_{\boldsymbol{X}}$ and $\ell \in \llbracket L \rrbracket, s \in [K]$. We also define, by the analogy with the main body, a family of plug-in estimators

$$\widehat{\pi}_{\boldsymbol{\Lambda},\mathbf{V}}(\ell \mid \boldsymbol{x}) \stackrel{\text{def}}{=} \sigma_\ell \left( \beta \left( \left\langle \boldsymbol{\lambda}_\ell - \boldsymbol{\nu}_\ell, \widehat{\boldsymbol{t}}(\boldsymbol{x}) \right\rangle - \widehat{r}_\ell(\boldsymbol{x}) \right)_{\ell=-L}^L \right) \qquad \boldsymbol{\Lambda}, \mathbf{V} \geqslant 0. \quad (33)$$

Our goal is to derive analogous optimization results for the new plug-in objective. Inspecting the proofs of Lemma 4.1 and Lemma 3.4, which bound variance of and the Lipschitz constant of the gradient respectively, we observe that those proofs only depend on the nature of $\widehat{\boldsymbol{t}}$ via Lemma F.4. In particular, the key quantity to control is

$$\widehat{\sigma}^2 = 2 \sum_{s \in [K]} \frac{\mathbb{E}_{\boldsymbol{X}}(p_s - \widehat{\tau}_s(\boldsymbol{X}))^2}{p_s^2}.$$

Before, when we assumed the perfect knowledge of $\tau$, the above was controlled by the Bhatia-Davis inequality, leveraging the fact that variance of $\tau_s(\boldsymbol{X})$ appears in the numerator. It is no longer the case here. However, if one can build calibrated estimators, that is, estimators for which $\mathbb{E}_{\boldsymbol{X}}[\widehat{\tau}_s(\boldsymbol{X})] = p_s$, the same machinery is applicable. In any case, even without requiring calibrated predictions, one can have a reasonable control of $\widehat{\sigma}^2$ building sufficiently accurate estimator $\widehat{\tau}_s$.

That being said, results of Lemma 4.1 and Lemma 3.4 generalize line-by-line, replacing $\sigma^2$ by $\widehat{\sigma}^2$ and give

$$\sup_{\boldsymbol{\Lambda},\mathbf{V} \geqslant 0} \mathbb{E}_{\boldsymbol{X}} \left\| \widehat{g}_{\boldsymbol{\Lambda},\mathbf{V}}(\boldsymbol{X}) - \nabla_{\boldsymbol{\Lambda},\mathbf{V}} \widehat{F}(\boldsymbol{\Lambda}, \mathbf{V}) \right\|^2 \leqslant \widehat{\sigma}^2 \quad \text{and} \quad \sup_{\boldsymbol{\Lambda},\mathbf{V}} \|\nabla^2 \widehat{F}(\boldsymbol{\Lambda}, \mathbf{V})\|_{\mathrm{op}} \leqslant \beta \widehat{\sigma}^2.$$

Moreover, the result of Lemma F.5 generalizes as well and gives $\sup_{\boldsymbol{\Lambda},\mathbf{V} \geqslant 0} \left\| \nabla_{\boldsymbol{\Lambda},\mathbf{V}} \widehat{F}(\boldsymbol{\Lambda}, \mathbf{V}) \right\| \leqslant \widehat{\sigma}$.

As in (19), we can show that

$$\left\| \left( -\nabla \widehat{F}(\boldsymbol{\Lambda}, \mathbf{V}) \right)_+ \right\| \leqslant \left\| \boldsymbol{G}_{\widehat{F},\alpha}(\boldsymbol{\Lambda}, \mathbf{V}) \right\| \qquad \forall \boldsymbol{\Lambda}, \mathbf{V} \geqslant 0, \quad (34)$$

where the gradient mapping $\boldsymbol{G}_{\widehat{F},\alpha}$ is defined by analogy with $\boldsymbol{G}_\alpha = \boldsymbol{G}_{F,\alpha}$, discussed in the main body.

Considering the SGD3 algorithm with the same choice of parameters, but replacing $\sigma^2$ by $\widehat{\sigma}^2$, results in a control of $\left\| \boldsymbol{G}_{\widehat{F},\alpha}(\widehat{\boldsymbol{\Lambda}}, \widehat{\mathbf{V}}) \right\|$.

*Proof of Lemma 5.1.* Fix $\boldsymbol{\Lambda}, \mathbf{V} \geqslant 0$. To ease the notation, we write $\widehat{\pi}$ to denote $\widehat{\pi}_{\boldsymbol{\Lambda}, \mathbf{V}}$ within this proof. Similarly to the proof of (21) from Lemma 3.5, one shows that for all $\boldsymbol{\Lambda}, \mathbf{V} \geqslant 0$

$$\sqrt{\sum_{\ell \in [\![L]\!] s \in [K]} \left( \left| \mathbb{E} \left[ \widehat{\pi}(\ell \mid \boldsymbol{X}) \widehat{t}_s(\boldsymbol{X}) \right] \right| - \varepsilon_s \right)_+^2} = \| (-\nabla \widehat{F}(\boldsymbol{\Lambda}, \mathbf{V}))_+ \| \qquad \forall \ell \in [\![L]\!], s \in [K].$$

Recalling that $\mathcal{U}_s(\widehat{\pi}, \ell) = |\mathbb{E}[\widehat{\pi}(\ell \mid \boldsymbol{X}) t_s(\boldsymbol{X})]|$, triangle's inequality combined with the above yields

$$\sqrt{\sum_{\ell \in [\![L]\!] s \in [K]} (\mathcal{U}_s(\widehat{\pi}, \ell) - \varepsilon_s)_+^2} \leqslant \| (-\nabla \widehat{F}(\boldsymbol{\Lambda}, \mathbf{V}))_+ \| + \sqrt{\sum_{\ell \in [\![L]\!] s \in [K]} \left\{ \mathbb{E}[\widehat{\pi}(\ell \mid \boldsymbol{X}) | \widehat{t}_s(\boldsymbol{X}) - t_s(\boldsymbol{X})|] \right\}^2}.$$

Cauchy-Schwartz inequality gives

$$\sum_{\ell \in [\![L]\!] s \in [K]} \left\{ \mathbb{E}[\widehat{\pi}(\ell \mid \boldsymbol{X}) | \widehat{t}_s(\boldsymbol{X}) - t_s(\boldsymbol{X})|] \right\}^2 \leqslant \sum_{s \in [K]} \mathbb{E} \left[ \left( \sum_{\ell \in [\![L]\!]} \widehat{\pi}(\ell \mid \boldsymbol{X})^2 \right) | \widehat{t}_s(\boldsymbol{X}) - t_s(\boldsymbol{X})|^2 \right].$$

Since $\sum_{\ell \in [\![L]\!]} \widehat{\pi}(\ell \mid \boldsymbol{X}) = 1$, then $\sum_{\ell \in [\![L]\!]} \widehat{\pi}(\ell \mid \boldsymbol{X})^2 \leqslant 1$. Thus, we have shown that

$$\sqrt{\sum_{\ell \in [\![L]\!] s \in [K]} (\mathcal{U}_s(\widehat{\pi}_{\boldsymbol{\Lambda}, \mathbf{V}}, \ell) - \varepsilon_s)_+^2} \leqslant \left\| \left( -\nabla \widehat{F}(\boldsymbol{\Lambda}, \mathbf{V}) \right)_+ \right\| + \mathbb{E}^{1/2} \| \widehat{\boldsymbol{t}}(\boldsymbol{X}) - \boldsymbol{t}(\boldsymbol{X}) \|^2.$$

We conclude using (34). ∎

*Proof of Lemma 5.2.* Fix $\boldsymbol{\Lambda}, \mathbf{V} \geqslant 0$. To ease the notation, we write $\widehat{\pi} \stackrel{\text{def}}{=} \widehat{\pi}_{\boldsymbol{\Lambda}, \mathbf{V}}$ and $\pi^\star \stackrel{\text{def}}{=} \pi_{\boldsymbol{\Lambda}^\star, \mathbf{V}^\star}$, within this proof.

As in the second part of the proof of Lemma 3.5, we have

$$\widehat{\mathcal{R}}_\beta(\widehat{\pi}) + \widehat{F}(\boldsymbol{\Lambda}, \mathbf{V}) \leqslant \left( \|(\boldsymbol{\Lambda}, \mathbf{V})\| + \alpha \widehat{\sigma} + \alpha \|\boldsymbol{\varepsilon}\| \sqrt{2(2L+1)} \right) \left\| \boldsymbol{G}_{\widehat{F}, \alpha}(\boldsymbol{\Lambda}, \mathbf{V}) \right\|. \qquad (35)$$

Furthermore, since $\| \nabla \operatorname{LSE}_\beta(\cdot) \|_1 \equiv 1$, we have

$$|\widehat{F}(\boldsymbol{\Lambda}, \mathbf{V}) - F(\boldsymbol{\Lambda}, \mathbf{V})| \leqslant \mathbb{E} \left[ \max_{\ell \in [\![L]\!]} \left\{ |r_\ell(\boldsymbol{X}) - \widehat{r}_\ell(\boldsymbol{X})| + \|\boldsymbol{\lambda}_\ell - \boldsymbol{\nu}_\ell\| \|\boldsymbol{t}(\boldsymbol{X}) - \widehat{\boldsymbol{t}}(\boldsymbol{X})\| \right\} \right],$$

and $|\widehat{\mathcal{R}}_\beta(\widehat{\pi}) - \mathcal{R}_\beta(\widehat{\pi})| \leqslant \mathbb{E}[\max_{\ell \in [\![L]\!]} \{|r_\ell(\boldsymbol{X}) - \widehat{r}_\ell(\boldsymbol{X})|\}]$. The last two displays combined with (35), gives

$$\mathcal{R}_\beta(\widehat{\pi}) + F(\boldsymbol{\Lambda}, \mathbf{V}) \leqslant \mathbb{E} \left[ 2 \max_{\ell \in [\![L]\!]} \left\{ |r_\ell(\boldsymbol{X}) - \widehat{r}_\ell(\boldsymbol{X})| + \|\boldsymbol{\lambda}_\ell - \boldsymbol{\nu}_\ell\| \|\boldsymbol{t}(\boldsymbol{X}) - \widehat{\boldsymbol{t}}(\boldsymbol{X})\| \right\} \right]$$
$$+ \left( \|(\boldsymbol{\Lambda}, \mathbf{V})\| + \alpha \widehat{\sigma} + \alpha \|\boldsymbol{\varepsilon}\| \sqrt{2(2L+1)} \right) \left\| \boldsymbol{G}_{\widehat{F}, \alpha}(\boldsymbol{\Lambda}, \mathbf{V}) \right\|.$$

Observe that $\min_{\boldsymbol{\Lambda}, \mathbf{V} \geqslant 0} F(\boldsymbol{\Lambda}, \mathbf{V}) = -\mathcal{R}_\beta(\pi^\star)$. Using triangle's inequality and the fact that $\max_{\ell \in [\![L]\!]} \|\boldsymbol{\lambda}_\ell - \boldsymbol{\nu}_\ell\| \leqslant \sqrt{2} \|(\boldsymbol{\Lambda}, \mathbf{V})\|$, we conclude recalling that $\mathcal{R}(\pi) + \frac{\log(2L+1)}{\beta} \geqslant \mathcal{R}_\beta(\pi) \geqslant \mathcal{R}(\pi)$ for any randomized prediction function. ∎

# F  Auxilliary results

In this appendix, we collect some standard auxiliary results, that are used to derive main claims of the paper.

**Lemma F.1** (Boyd and Vandenberghe (2004)). *It holds that*

$$\mathrm{LSE}_\beta(\boldsymbol{w}) = \max_{\boldsymbol{p} \in \Delta} \left\{ \langle \boldsymbol{w}, \boldsymbol{p} \rangle - \frac{1}{\beta} \Psi(\boldsymbol{p}) \right\} ,$$

*where $\Delta$ is the probability simplex in $\mathbb{R}^m$ and $\Psi(\boldsymbol{p}) = \sum_{i=1}^m p_i \log(p_i)$. Furthermore, $-\Psi(\boldsymbol{p}) \in [0, \log(m)]$ and the optimum in the above optimization problem is achieved at $\boldsymbol{p}^\star = \sigma(\beta \boldsymbol{w})$.*

**Lemma F.2** (Gao and Pavel (2017)). *Let $\boldsymbol{a} = (a_1, \cdots, a_m)$ and $\beta > 0$. Define log-sum-exp and softmax functions respectively as*

$$\mathrm{LSE}_\beta(\boldsymbol{a}) \stackrel{\mathrm{def}}{=} \frac{1}{\beta} \log \left( \sum_{i=1}^m \exp(\beta a_i) \right) \text{ and } \sigma_j(\beta \boldsymbol{a}) \stackrel{\mathrm{def}}{=} \frac{\exp(\beta a_j)}{\sum_{i=1}^m \exp(\beta a_i)} \quad j \in [m] .$$

*The LSE property is as follows*

$$\max\{a_1, \cdots, a_m\} \leqslant \mathrm{LSE}_\beta(\boldsymbol{a}) \leqslant \max\{a_1, \cdots, a_m\} + \frac{\log(m)}{\beta} .$$

*Moreover, $\sigma(\beta \boldsymbol{a}) = \nabla \mathrm{LSE}_\beta(\boldsymbol{a})$, and $\sigma(\beta \boldsymbol{a})$ is $\beta$-Lipschitz.*

**Lemma F.3** (Bhatia and Davis (2000)). *Let $m$ and $M$ be the lower and upper bounds, respectively, for a set of real numbers $a_1, \cdots, a_n$, with a particular probability distribution. Let $\mu$ and $\sigma^2$ be respectively the expected value and the variance of this distribution. Then the Bhatia–Davis inequality states:*

$$\sigma^2 \leqslant (M - \mu)(\mu - m) .$$

**Lemma F.4.** *It holds that*

$$\mathbb{E}_{\boldsymbol{X}} \left[ \sum_{s \in [K]} t_s^2(\boldsymbol{X}) \right] \leqslant \sum_{s \in [K]} \frac{1 - p_s}{p_s} ,$$

*where $t_s(x) = 1 - \frac{\tau_s(x)}{p_s}$.*

*Proof.* We have $\mathbb{E}_{\boldsymbol{X}}[\tau_s(\boldsymbol{X})] = p_s$ and $0 \leqslant \tau_s(\boldsymbol{X}) \leqslant 1$ almost surely. Using Bhatia-Davis inequality written in Lemma F.3, we deduce that

$$\mathbb{E}_{\boldsymbol{X}} \left[ \sum_{s \in [K]} t_s^2(\boldsymbol{X}) \right] = \sum_{s \in [K]} \mathrm{Var} \left( \frac{\tau_s(\boldsymbol{X})}{p_s} \right) = \sum_{s \in [K]} \frac{1}{p_s^2} \mathrm{Var}\left(\tau_s(\boldsymbol{X})\right) \leqslant \sum_{s \in [K]} \frac{1 - p_s}{p_s} .$$

The proof is concluded. ∎

**Lemma F.5.** *Let $\sigma^2 \stackrel{\mathrm{def}}{=} 2 \sum_{s \in [K]} \frac{1 - p_s}{p_s}$. It holds that $\|\nabla_{\boldsymbol{\Lambda}, \mathbf{V}} F(\boldsymbol{\Lambda}, \mathbf{V})\| \leqslant \sigma + \sqrt{2(2L + 1)} \|\boldsymbol{\varepsilon}\|$.*

*Proof.* By Jensen's inequality

$$\|\nabla_{\boldsymbol{\Lambda}, \mathbf{V}} F(\boldsymbol{\Lambda}, \mathbf{V})\|^2 = \|\mathbb{E}_{\boldsymbol{X}}[g_{\boldsymbol{\Lambda}, \mathbf{V}}(\boldsymbol{X})]\|^2 \leqslant \mathbb{E}_{\boldsymbol{X}}[\|g_{\boldsymbol{\Lambda}, \mathbf{V}}(\boldsymbol{X})\|^2] .$$

Recalling the definition of $g_{\boldsymbol{\Lambda}, \mathbf{V}}$, given in (11), we have

$$\mathbb{E}_{\boldsymbol{X}}[\|g_{\boldsymbol{\Lambda}, \mathbf{V}}(\boldsymbol{X})\|^2] = \mathbb{E}_{\boldsymbol{X}} \sum_{\ell \in [\![L]\!] s \in [K]} \left( (\sigma_\ell(\cdot) t_s(\boldsymbol{X}) + \varepsilon_s)^2 + (-\sigma_\ell(\cdot) t_s(\boldsymbol{X}) + \varepsilon_s)^2 \right)$$

$$= 2 \mathbb{E}_{\boldsymbol{X}} \sum_{\ell \in [\![L]\!] s \in [K]} \left( \sigma_\ell^2(\cdot) t_s^2(\boldsymbol{X}) + \varepsilon_s^2 \right) \leqslant \sigma^2 + 2(2L + 1) \|\boldsymbol{\varepsilon}\|^2 ,$$

where the last inequality comes from the proof of Lemma 4.1.

The proof is concluded. ∎

## G   Additional details on experiments

**Evaluation measures.**    We use $\mathcal{D}_{\text{test}} = \{(\boldsymbol{x}'_i, s'_i, y'_i)\}_{i=1}^{m}$ to collect the following statistics of any (randomized) prediction $\pi$

$$\widehat{\mathcal{R}}(\pi) \stackrel{\text{def}}{=} \frac{1}{m} \sum_{i=1}^{m} \int_{-\infty}^{+\infty} (\widehat{y} - y'_i)^2 \, \pi(\mathrm{d}\,\widehat{y} \mid \boldsymbol{x}'_i),$$

$$\widehat{U}_s(\pi) \stackrel{\text{def}}{=} \sup_{t \in \mathbb{R}} \left| \frac{1}{m_s} \sum_{i=1}^{m} \int_{-\infty}^{t} \pi(\mathrm{d}\,\widehat{y} \mid \boldsymbol{x}'_i) \mathbb{I}\{s'_i = s\} - \frac{1}{m} \sum_{i=1}^{m} \int_{-\infty}^{t} \pi(\mathrm{d}\,\widehat{y} \mid \boldsymbol{x}'_i) \right|,$$

which correspond to the empirical risk and the empirical group-wise unfairness quantified by the Kolmogorov-Smirnov distance of a randomized. We note that our classifier is supported on a finite grid, thus all the integrals involved transform into weighted sums.

Agarwal et al. (2019) build multi-class classifiers $h_k : \mathbb{R}^d \mapsto \Theta$, where $\Theta$ is some finite grid over $\mathbb{R}$ and $k = 1, \ldots, K$, that come with weights $(w_1, \ldots, w_K)^\top$ such that $w_k \geqslant 0$ and $\sum_{k=1}^{K} w_k = 1$. Then, they build a randomized classifier $\pi(\cdot \mid \cdot)$ such that $\operatorname{supp}(\pi(\cdot \mid \boldsymbol{x})) = \Theta$ and for each $\theta \in \Theta$

$$\mathbb{P}(\widehat{Y}_\pi = \theta \mid \boldsymbol{X} = \boldsymbol{x}) = \sum_{k=1}^{K} w_k \mathbb{I}\{h_k(\boldsymbol{x}) = \theta\}.$$

Thus, integrals appearing in $\widehat{\mathcal{R}}$ and $\widehat{U}_s$ reduced to finite sums for both methods.

**Additional details on the experiments on *Communities and Crime* and *Law School* datasets.**
*Communities and Crime* dataset has 1994 instances, however we use 1984 examples with 120 features after preprocessing. *Law School* dataset has 20649 instances, thus we use a smaller sub-sample of 2000 points with 11 features after preprocessing.

We take the sets of unfairness thresholds $\{(2^{-i}, 2^{-i})_{i \in \mathcal{I}}\}$, where $\mathcal{I} = \{1, 2, 4, 5, 6, 8, 16, 32, 128, 512\}$ for *Communities and Crime* dataset, and $\mathcal{I} = \{0, 1, 2, 4, 5, 6, 8, 16, 32, 64, 128\}$ for *Law School* dataset. We train *Communities and Crime* dataset for N=15000 iterations and *Law School* dataset for N=5000 iterations for each pair of epsilons. We use parameters $L = \sqrt{T}$, $\beta = \sqrt{T} \log \sqrt{T}$ and $B = 1$ for both datasets. We repeat the aforementioned pipeline 10 times to ensure more reliable statistical summary.

**Discussion on other algorithms.**    We conduct additional experiments to observe the behaviors of the more straightforward algorithms discussed in Appendix C. We illustrate the comparison in Figure 3. In conclusion, all of the algorithms perform similarly in the middle to high unfairness regime, while those based on `SGD3` are more stable in the low unfairness (high fairness) regime.

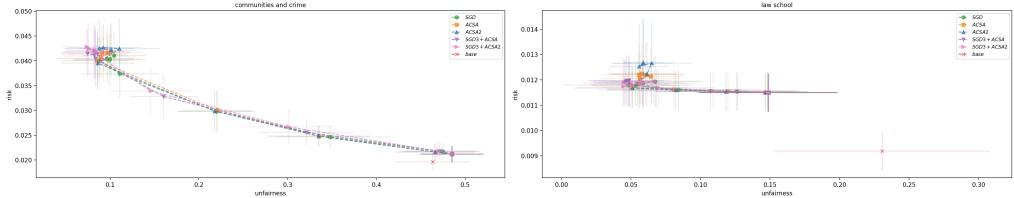

Figure 3: Comparison of `SDG`, `ACSA`, `ACSA2`, `SDG3+ACSA` and `SDG3+ACSA2` algorithms on *Communitites and Crime* and *Law School* datasets.

**Additional experiments on *Adult* dataset.**    We conduct further experiments on *Adult* dataset (Lichman (2013)). Classically, *Adult* is used for classification, however we use it to predict individual's age on a scale of 0 to 100, normalized to $[0, 1]$. We factor in sex as a sensitive attribute, distinguishing between male and female individuals. *Adult* dataset has 48842 instances, however we clean and preprocess it, and use a smaller sub-sample of 2000 points with 8 features throughout our experiments.

The pipleline of the experiments is the same as the one for *Law School* and *Communities and Crime* datasets in the main body. We randomly split the data into training, unlabeled and testing sets

|              | DP-postproc     | ADW-1              | ADW-2             |
| ------------ | --------------- | ------------------ | ----------------- |
| communities  | $5.89 \pm 0.47$ | $378.39 \pm 263.77$ | $199.05 \pm 161.18$ |
| law school   | $0.78 \pm 0.08$ | $240.53 \pm 178.68$ | $136.3 \pm 96.73$ |
| adult        | $3.7 \pm 0.34$  | $174.57 \pm 91.7$  | $96.78 \pm 61.35$ |

Table 1: The average training time (in seconds) for one $\varepsilon$ threshold.

with proportions of $0.4 \times 0.4 \times 0.2$. We use $\mathcal{D}_{\text{train}} = \{(\boldsymbol{x}_i, s_i, y_i)_{i=1}^n\}$ to train a base (unfair) regressor to estimate $\eta$ and to train a classifier to estimate $\tau$. We use simple *LinearRegression* and *LogisticRegression* from *scikit-learn* for training the regressor and the classifier, and give them to Algorithm 1 with $\mathcal{D}_{\text{unlabeled}} = (\boldsymbol{x})_{i=n+1}^{n+T}$ for $N = 10000$ iterations. We use $\mathcal{D}_{\text{test}} = \{(\boldsymbol{x}_i', s_i', y_i')\}_{i=1}^m$ to collect statistics. We take the sets of unfairness thresholds $\{(2^{-i}, 2^{-i})_{i \in \mathcal{I}}\}$, where $\mathcal{I} = \{0, 1, 2, 4, 5, 6, 8, 16, 32, 64, 128\}$ for. As in the experiments in the main body, we set $L = \sqrt{T}$, $\beta = \sqrt{T}/\log\sqrt{T}$ and $B = 1$. We repeat the pipeline 10 times.

We compare our method with the ADW method. We train ADW 2 times: we use $\mathcal{D}_{\text{train}}$ and $\mathcal{D}_{\text{unlabeled}}$ as training set for ADW-1, whereas for ADW-2 we use only $\mathcal{D}_{\text{train}}$. We take the set $\{(2^{-i}, 2^{-i})_{i \in \mathcal{I}}\}$, where $\mathcal{I} = \{1, 2, 4, 8, 16\}$ as unfairness thresholds for training both datasets. We train ADW-1 and ADW-2 for each pair of epsilons for 10 times.

In Figure 4 we illustrate the convergence of the risk and the unfairness for convergence for $\varepsilon = (2^{-8}, 2^{-8})$ unfairness threshold. We also illustrate the comparison of our model with ADW-1, ADW-2 and base models.

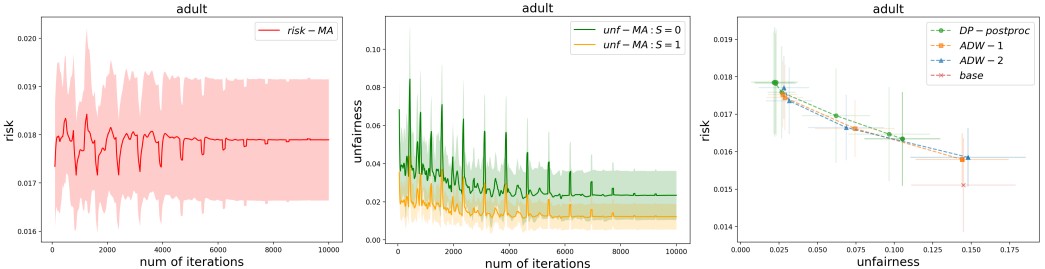

Figure 4: Experiment on *Adult* dataset: risk convergence, unfairness convergence and comparison with ADW.

**Running time.** Additional details about training time for *Communities and Crime*, *Law School* and *Adult* datasets are presented in Table 1.

**Additional experiments on a synthetic dataset.** We conduct an additional experiment to demonstrate the results of Algorithm 1 in the case of multiple sensitive attributes. We generate a synthetic dataset $\mathcal{D}_n = (\boldsymbol{X}_i, S_i, y_i)_{i=1}^n$ of $n = 2000$ points, where $(\boldsymbol{X}_i)_i^n = (X_{i1}, X_{i2}, X_{i2})_i^n \sim \mathcal{N}(0, 1)$. We choose $S_i = 0$ if $X_{i1} \leqslant -0.7$, $S_i = 1$ if $X_{i1} < 0$, $S_i = 2$ if $X_{i1} < 0.7$ and $S_i = 4$ if $X_{i1} \geqslant -0.7$. For $i \in [n]$, we generate $y_i = 4\sum_{j=1}^3 X_{ij} + X_{i1} + \xi_i$, where $\boldsymbol{\xi} = (\xi_i)_i^n \sim \mathcal{N}(0, 1)$. We split $\mathcal{D}_n$ into *train*, *unlabeled* and *test* datasets with proportions of $0.4 \times 0.4 \times 0.2$. We use $(\boldsymbol{X}_{train}, \boldsymbol{y}_{train})$ to train the base estimator, $(\boldsymbol{X}_{train}, \boldsymbol{S}_{train})$ to train the classifier and $\boldsymbol{X}_{unlab}$ to train the fair regression model. We evaluate our model on *test* dataset. In Figure 5 we illustrate the distributions of the predictions (scaled to $[-1, 1]$) of the fair and base models.

This experiment is for visual representation of the Algorithm 1 in the case of multiple sensitive attributes, thus we do not collect further statistics.

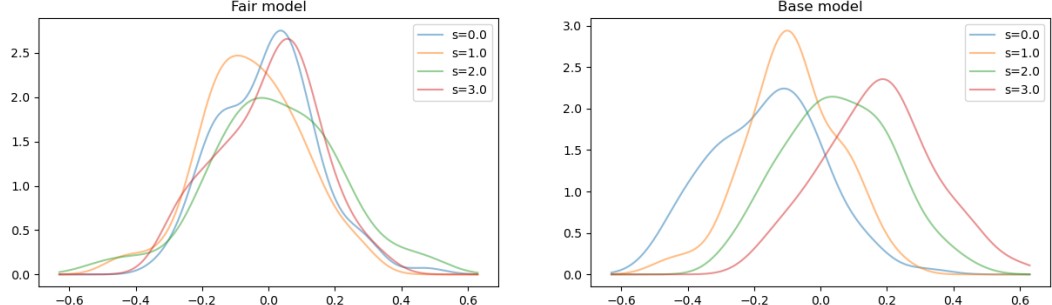

Figure 5: The distributions of the (scaled) predictions of the fair and base models.

