# OpenReview forum: "Regression under demographic parity constraints via unlabeled post-processing"
_NeurIPS.cc/2024/Conference — NeurIPS 2024 poster_

### Official Review · Reviewer_qMMa · 2024-06-25

**Soundness:** 3
**Presentation:** 3
**Contribution:** 3
**Rating:** 7
**Confidence:** 4

**Summary:**

This paper considers post-processing regressors to satisfy the group fairness criterion of demographic parity, in particular, under the attribute unaware setting.

1. The authors begin by analyzing the constrained optimization problem (\*) for fair post-processing, and showing in lemma 3.1 that the solution, i.e., the post-processed regressor, can be represented and parameterized by the dual variables of (\*); and provided that the base models are accurate and (\*) is solved to optimiality, the resulting regressor will be optimally fair.

2. Then the authors discussed optimization and statistical aspects of (\*). Specifically, the convergence rate is analyzed, as well as the conditions for non-perfect solutions to satisfy fairness, namely that the "gradient mapping" of (\*) needs to be lipschitz. For this reason, the authors recommended using the SGD3 algorithm for optimizing (\*).

3. The paper closes with empirical evaluations of the proposed algorithm.

**Strengths:**

1. To my knowledge, this is the first paper that studies fair post-processing regressors in the attribute unaware setting.
2. The primal-dual analysis that leads to the representation result in lemma 3.1 is interesting. Because the proposed procedure involves discretization, the formulation of (\*) is based on the support of the regressor's output space. This is different from somewhat similar works for the classification setting [1, 2] where the optimization problem is based on the scores of the training examples (without discretization).
3. The theoretical analysis is thorough, hence the proposed procedure is well-supported, including the choice of SGD3.

[1] https://arxiv.org/pdf/2310.05725
[2] https://arxiv.org/pdf/2405.04025

**Weaknesses:**

1. Regarding lipschitzness of $F$. The reviewer feels that some discussions are curt. The authors introduced the "gradient mapping" $G_\alpha$ in eq. (8), with a hyperparameter $\alpha$. It is not mentioned: how $\alpha$ should be chosen, in theory or in practice; how exactly SGD3 controls the norm of $G_\alpha$ (lines 237-239) in the main body. At least a brief discussion should be included in the main body for the latter, since theorem 5.1 would depend on whether SGD3 can provably reduce this quantity.
2. Regarding the use of SGD3, an ablation study would have been appreciated at illustrating the importance of using SGD3 over other optimizers; is it absolutely necessary? Also, could the authors discuss potential limitations with SGD3?
3. The authors mentioned that there are several hyperparameters associated with SGD3 (line 236), and with discretization (i.e., the number of bins). But it does not seem to be discussed how these are chosen for the experiments in section 6, and whether mis-specification of these hyperparameters would impact performance.

[3] https://arxiv.org/pdf/2006.07286

**Questions:**

- Does the analysis rely on properties unique to the demographic parity fairness criterion? If not, is there a path to extend to other criteria (potentially taking inspirations from [1, 2])?

**Limitations:**

- See weaknesses.
- Minor, but the proposed procedure requires discretizing the support, which is a limitation; even though the reviewer is aware that, in practice, discretization helps with generalization hence gives better performing models compared to non-parametric methods [3].

---

> ### Author Rebuttal · Authors · 2024-08-05
>
> We appreciate the reviewer's careful reading and fully agree with the evaluation regarding the strengths and potential directions for future investigation. We will address the questions raised below.
>
> *Weaknesses*
>
> **W1:**
>
> * Regarding lipschitzness of $F$. The reviewer feels that some discussions are curt.
>
> **A:** We will enhance our discussion to better emphasize that the smoothness properties of $F$ are controlled by the regularization parameter $\beta$ *(Lemma 3.4)*. Our final theoretical guarantees also provide practical guidance on selecting this parameter, balancing the regularization error *(Lemma 3.3)* and ensuring the algorithm's fast convergence.
>
> * The authors introduced the "gradient mapping" $G_\alpha$ in eq. (8), with a hyperparameter $\alpha$. It is not mentioned how $\alpha$ should be chosen, in theory or in practice.
>
> **A:** In theory, we set $\alpha = 1/M$ and any value of $\alpha \leq 1/M$ would yield exactly the same result. However, the choice of $\alpha$ is not critical for our specific problem, as the bounds on unfairness and risk are valid for any $\alpha > 0$ (see *Lemma 3.5*, lines 203-204). In practice, there is no $\alpha$ hyperparameter in the final implementation of SGD3 with a black-box optimizer.
>
> *  It is not mentioned how exactly SGD3 controls the norm of $G_\alpha$ in the main body.
>
> **A:** Due to space limitations, we have provided the details of this part in the *Appendix C*. We would also like to remind that any algorithm designed to minimize the expected squared norm of the gradient mapping can be used. SGD3 is just an example that we included because it is supported by convergence rates.
>
> **W2:**
> * An ablation study would have been appreciated at illustrating the importance of using SGD3 over other optimizers; is it absolutely necessary?
>
> **A:** We conducted additional experiments to observe the behaviors of simpler algorithms. A figure that illustrates the comparison is included in the attached pdf file. In conclusion, all algorithms perform similarly in the middle to high unfairness regime, while those based on SGD3 are more stable in the low unfairness (high fairness) regime. However, since the stochastic minimization of the norm of the gradient of a convex function is a relatively niche topic, there are only a few algorithms with established convergence rates. As we aim for end-to-end guarantees, we have chosen methods based on SGD3. Nonetheless, other stochastic minimization methods could potentially perform just as well in practice.
>
> * Also, could the authors discuss potential limitations with SGD3?
>
> **A:**  The SGD3 algorithm, as described by [Allen-Zhu (2021)](https://arxiv.org/pdf/1801.02982), serves primarily as a theoretical model that highlights a unique phenomenon in stochastic convex optimization with an unconventional criterion. Between the submission and the rebuttal, we extended and simplified the theory from [Foster et al. (2019)](https://arxiv.org/pdf/1902.04686) to be applicable to our problem. This has enabled us to provide a theoretical guarantee for a simpler algorithm that combines an SGD3-like approach with accelerated stochastic gradient descent. While this improved analysis slightly enhances our fairness and risk guarantees, it does not alter the main message of the paper.
>
> **W3:** The authors mentioned that there are several hyperparameters associated with SGD3 (line 236), and with discretization (i.e., the number of bins). But it does not seem to be discussed how these are chosen for the experiments in section 6, and whether mis-specification of these hyperparameters would impact performance.
>
> **A:** In practice, we normalize the target variables to the range $[-1, 1]$, so we set $B = 1$ as our default practical recommendation, which aligns with actual practice. We have found that the other hyperparameters suggested by our theory already produce good empirical results. The exact choices of these parameters are detailed in *Appendix G* (line 693), and, aside from some multiplicative constants, they closely follow the theoretical values. These will be the default settings in the package we plan to release.
>
> *Questions*
>
> **Q:** Does the analysis rely on properties unique to the demographic parity fairness criterion? If not, is there a path to extend to other criteria (potentially taking inspirations from [1, 2])?
>
> **A:** This is an excellent question. As the reviewer may have noticed, the key feature that ensures everything works is the compatibility of the demographic parity constraint with discretization. This approach can be adapted to any other notion of fairness that is "friendly" to discretization. However, care must be taken when introducing new notions of fairness. The demographic parity constraint is convenient because there are always predictions that satisfy it (e.g., constants), but this may not be true for other fairness notions. Since our algorithm relies heavily on a primal-dual approach, the finiteness of the optimal dual variables can only be guaranteed if certain constraint qualification conditions are met. This is not an issue for the demographic parity constraint but could be a potential obstacle for other fairness definitions.
>
> We appreciate the reviewer's suggestion to include those two references, and they will be added to the final version. We want to emphasize that both references heavily depend on the ability to express the form of the optimal classifier under a given fairness constraint analytically. However, this is not applicable to regression in the unawareness setup as no such formula exists. Nevertheless, combining our discretization approach with the two references could potentially lead to a sensible method, and it is indeed an interesting avenue for future research.

---

> > ### Comment · Reviewer_qMMa · 2024-08-11
> >
> > The reviewer thanks the authors for the response.
> >
> > Could the authors also (briefly) comment on how the number of bins is chosen (in the experiments), and how it affects performance (theoretically)?

---

> > > ### Author Response · Authors · 2024-08-12
> > >
> > > We appreciate the reviewer's feedback. In Lemma B.1 (Appendix B, line 518), we show that discretization increases the risk by $4B/L + 1/L^2 + \log(2L+1)/\beta$. Additionally, as demonstrated in the proof of Theorem 5.1 (Appendix D, line 615), setting $L = \sqrt{T}$ achieves a risk rate of $1/\sqrt{T}$. Therefore, we use $L = \sqrt{T}$ in our experiments, resulting in $2\sqrt{T} + 1$ bins, where T is the number of unlabeled samples. We also note that using more bins wouldn't improve statistical performance but would complicate the optimization due to higher dimensionality. Hence, we adhere to the theoretically recommended setting in practice.

---

> > > > ### Comment · Reviewer_qMMa · 2024-08-13
> > > >
> > > > The reviewer thanks the authors for the response, and think the readers would appreciate if these points are highlighted in the main body.  The reviewer has raised their score.

---

### Official Review · Reviewer_RqJN · 2024-07-10

**Soundness:** 4
**Presentation:** 3
**Contribution:** 3
**Rating:** 7
**Confidence:** 4

**Summary:**

This paper proposes an algorithm that takes in an fitted regression function and a sensitive attribute predictor and output a prediction function satisfying the demographic parity constraint. It designs a smooth convex objective function with discretization to solve for a prediction function with small risk and controlled violation of the demographic parity constraint. Stochastic minimization techniques are applied to solve the proposed optimization problem and recover the statistical rate $1/\sqrt{T}$.

**Strengths:**

The proposed algorithm is supported by theoretical analysis and error bounds. The proofs are well-organized and clear to read. The algorithm deploys suitable stochastic minimization techniques to achieve statistical guarantees. Moreover, based on the experiment results, the proposed algorithm is much more computationally efficient than existing methods.

**Weaknesses:**

Typos:
* falls withing -> falls within (page 2)
* statisitcal properties -> statistical properties (page 5)
* out approach -> our approach (page 7)
* phenomenons -> phenomena (page 7)
* outperfomce -> outperform (page 8)

**Questions:**

In practice, how do we pick algorithm parameters such as L, B and \beta?

**Limitations:**

Yes.

---

> ### Author Rebuttal · Authors · 2024-08-05
>
> Thank you to the reviewer for the careful reading and evaluation. We agree with the feedback and will correct the suggested typos in the revision. We will also address the question raised below.
>
> **Q**: In practice, how do we pick algorithm parameters such as L, B and $\beta$?
>
> **A**:  In practice, we normalize the target variables to the range $[-1, 1]$, so we set $B = 1$ as our default practical recommendation, which aligns with actual practice. We have found that the other hyperparameters suggested by our theory already produce good empirical results. The exact choices of these parameters are detailed in Appendix G (line 693), and, aside from some multiplicative constants, they closely follow the theoretical values. These will be the default settings in the package we plan to release.

---

> > ### Comment · Reviewer_RqJN · 2024-08-13
> >
> > I thank the authors for their elaborate rebuttal. This does address my concerns and questions.

---

### Official Review · Reviewer_3wik · 2024-07-11

**Soundness:** 1
**Presentation:** 2
**Contribution:** 2
**Rating:** 5
**Confidence:** 3

**Summary:**

This paper presents a post-processing algorithm designed to enforce demographic parity in regression tasks without access to sensitive attributes during inference.

**Strengths:**

The solution is versatile solution for enforcing demographic parity as it can be applied on different optimizers.

The paper provides a rigorous theoretical foundation to quantify the risk.

The fairness problem is relevant.

**Weaknesses:**

While the algorithm presented in the paper appears promising, its evaluation is hampered by the presentation style, which tends to obscure clarity. Technical details such as the reliance on discretization and sophisticated methods for controlling the gradient norm are emphasized in the abstract, which could be streamlined to focus on broader impacts and significance instead.

The paper contains numerous remarks that disrupt the flow of discussion; for instance, Remark 2.1 seems tangential and could be relegated to an appendix or omitted entirely. Other generalizations and discussions, such as those in Remark 2.2 and the paragraph on line 304, should be consolidated and presented at the end of the paper to maintain focus. The 'abuses notation' in Remark 3.1 could be resolved by providing clear definitions Comparisons with other literature are inconsistently integrated within the text, appearing at disparate locations such as lines 56, 228, and 334, which could be better organized to aid in comparative analysis and enhance readability.

There is excessive use of sub-titles or mini-sections. Some of these sections are notably brief (line 176, for example), where the short content under each title does not justify the need for a separate heading.

**Questions:**

The paper focuses on demographic parity as a fairness metric. Are there impacts on other fairness metrics like Equalized Odds and Equal Opportunity when using the proposed algorithm? Does it improve these metrics, or could it potentially worsen them?

Regarding the risks plot in Figure 1, could you clarify its purpose given there are no comparative benchmarks provided? How should the unfairness score presented in the plot be interpreted?

Reference [1] addresses a similar topic with a minimax approach. Is this minimax result applicable or relevant to the methodology used in your paper?

In Algorithm 1, "DP" is mentioned in the name. Could you specify what "DP" stands for?

The algorithm "fairname" appears in line 305 without a prior definition. Could you explain what this term means within the context of your study?

[1]  Fukuchi, Kazuto, and Jun Sakuma. "Demographic parity constrained minimax optimal regression under linear model." Advances in Neural Information Processing Systems 36 (2024).

**Limitations:**

Not applicable.

---

> ### Author Rebuttal · Authors · 2024-08-05
>
> We appreciate the reviewer's time and effort in reviewing our work, but we respectfully disagree  with the evaluation.
>
> *Weaknesses*
>
> The reviewer has criticized several stylistic choices in our paper. While we recognize that these choices may not align with the reviewer's preferences, it is worth noting that neither of the other two reviewers raised similar concerns. Our approach is driven by theory and is based on a carefully developed methodology rooted in mathematical reasoning, rather than common sense. Given this theoretical foundation, we believe our stylistic choices are justified and aid the reader in understanding our thought process. Below, we address the specific stylistic comments raised:
>
>
> 1.  *"Remark 2.1"*. We disagree with the assessment that *Remark 2.1* is tangential. On the contrary, clearly defining the joint distribution of every random variable involved is crucial for understanding the problem. Including *Remark 2.1* helps prevent potential misconceptions, such as assuming that $\hat{Y} = Y$ is a valid prediction function for any distribution of $(X, S, Y)$. *Remark 2.1* explicitly clarifies that $\hat{Y} = Y$ is not a valid prediction function unless $Y$ is measurable with respect to $X$.
>
> 2. *"Remark 2.2"*. *Remark 2.2* anticipates potential questions from readers about possible extensions of our methodology. While we do not insist on keeping this remark in the main body, we believe it could be useful for some readers.
>
> 3. *"Remark 3.1"*.  We do not see any issue in abuse of notation as it is extremely common in mathematical science. Yet, we find it equally important to be extremely clear when such an abuse happens so that the reader is aware and could easily understand the logic behind our choice. The reviewer suggests providing clear definitions, but we would appreciate more specific guidance: which quantities are not clearly defined in our paper?
>
> 4. *"There is excessive use of sub-titles or mini-sections"*. This is a stylistic choice we made to clarify the purpose of each paragraph. We believe it helps in setting the context clearly. We prefer to keep this as is unless the reviewer can provide a clearly better alternative.
>
> *Questions*
>
> **Q1:** The paper focuses on demographic parity as a fairness metric. Are there impacts on other fairness metrics like Equalized Odds and Equal Opportunity when using the proposed algorithm? Does it improve these metrics, or could it potentially worsen them?
>
> **A1:** Equalized Odds and Equal Opportunity typically refer to fairness conditions in binary classification, while our work addresses fairness in the context of regression. While extensions of these notions to regression are possible, our approach focuses on demographic parity. As such, it might improve, worsen, or maintain fairness under other definitions. Since we have not claimed to address all definitions of fairness simultaneously, studying the trade-offs between different fairness notions is beyond the scope of this work. However, we will include a discussion emphasizing that our algorithm is tailored for a specific fairness definition as it is highlighted by the title.
>
> **Q2:** Regarding the risks plot in *Figure 1*, could you clarify its purpose given there are no comparative benchmarks provided? How should the unfairness score presented in the plot be interpreted?
>
> **A2:** The purpose of *Figure 1* is to illustrate the post-processing dynamics of our proposed method, not to compare it with other algorithms, which are based on different approaches. A comparison with other algorithms is shown in *Figure 2*. Additional details on the implementation are provided in *Appendix G*, and the code is available via the link provided in the paper.
>
> **Q3:** Reference [1] addresses a similar topic with a minimax approach. Is this minimax result applicable or relevant to the methodology used in your paper?
>
> **A3:**  The paper suggested by the reviewer has little in common with our contribution. Firstly, it deals with the scenario where sensitive attributes are available for prediction (the awareness setup) and relies on an explicit form of the optimal prediction provided by [Chzhen et al. (2020b)](https://arxiv.org/pdf/2006.07286) and by [Le Gouic et al. (2020)](https://arxiv.org/pdf/2005.11720). Such an explicit form is not yet available for regression in the unawareness setup, which remains an open question in the field. Secondly, that paper focuses exclusively on linear regression and provides a minimax statistical analysis for this case. In contrast, our work presents a versatile, theoretically grounded post-processing algorithm that can be applied to any pre-trained model in the unawareness setup for regression.
>
> **Q4:** In Algorithm 1, "DP" is mentioned in the name. Could you specify what "DP" stands for?
>
> **A4:** DP in the name of *Algorithm 1* "DP post-processing" stands for *demographic parity*. We have indeed not introduced this term in the main body and will include it in the revision.
>
> **Q5:** The algorithm "fairname" appears in line 305 without a prior definition. Could you explain what this term means within the context of your study?
>
> **A5:** The name "fairname" in $\texttt{FairName}(L, T, \beta, \boldsymbol{p}, B, \hat\eta, \hat{\boldsymbol{\tau}})$ in the section of extension to unknown $\eta$ and $\tau$ is an unfortunate typo. It should be  $\texttt{DP post-processing}$ and it will be corrected upon revision.

---

> > ### Comment · Reviewer_3wik · 2024-08-13
> >
> > Thank you for the the response, which has addressed most of my concerns. I believe the authors will improve the readability in the final version. I am no longer opposed to the acceptance of this paper and will adjust my score accordingly.

---

### Author Rebuttal · Authors · 2024-08-05

We conducted an ablation study to observe the behaviors of other algorithms, as suggested by Reviewer qMMa. A figure that illustrates the comparison is included in the attached pdf file.

In conclusion, all algorithms perform similarly in the middle to high unfairness regime, while those based on SGD3 are more stable in the low unfairness (high fairness) regime. However, since the stochastic minimization of the norm of the gradient of a convex function is a relatively niche topic, there are only a few algorithms with established convergence rates. As we aim for end-to-end guarantees, we have chosen methods based on SGD3. Nonetheless, other stochastic minimization methods could potentially perform just as well in practice.

---

### Decision · Program_Chairs · 2024-09-25

**Decision:**

Accept (poster)

**Comment:**

The paper studies the problem of fair regression with demographic parity constraints under a setting where sensitive features are not revealed at the test time. The setting is new and well-motivated. The paper develops algorithms with theoretical guarantees and is supported with numerical experiments.